# Multi-ancestry genome-wide meta-analysis identifies novel basal cell carcinoma loci and shared genetic effects with squamous cell carcinoma

Hélène Choquet [1✉], Chen Jiang[1], Jie Yin [1], Yuhree Kim[2,3], Thomas J. Hoffmann [4,5], 23andMe Research Team*, Eric Jorgenson [6] & Maryam M. Asgari[2,3]

Basal cell carcinoma (BCC) is one of the most common malignancies worldwide, yet its genetic determinants are incompletely defined. We perform a European ancestry genome-wide association (GWA) meta-analysis and a Hispanic/Latino ancestry GWA meta-analysis and meta-analyze both in a multi-ancestry GWAS meta-analysis of BCC, totaling 50,531 BCC cases and 762,234 controls from four cohorts (GERA, Mass-General Brigham Biobank, UK Biobank, and 23andMe research cohort). Here we identify 122 BCC-associated loci, of which 36 were novel, and subsequently fine-mapped these associations. We also identify an association of the well-known pigment gene *SLC45A2* as well as associations at *RCC2* and *CLPTM1L* with BCC in Hispanic/Latinos. We examine these BCC loci for association with cutaneous squamous cell carcinoma (cSCC) in 16,407 SCC cases and 762,486 controls of European ancestry, and 33 SNPs show evidence of association. Our study findings provide important insights into the genetic basis of BCC and cSCC susceptibility.

[1] Kaiser Permanente Northern California (KPNC), Division of Research, Oakland, CA, USA. [2] Department of Dermatology, Massachusetts General Hospital, Boston, MA, USA. [3] Department of Population Medicine, Harvard Medical School and Harvard Pilgrim Health Care Institute, Boston, MA, USA. [4] Institute for Human Genetics, University of California, San Francisco (UCSF), San Francisco, CA, USA. [5] Department of Epidemiology and Biostatistics, UCSF, San Francisco, CA, USA. [6] Regeneron Genetics Center, Tarrytown, NY, USA. *A list of authors and their affiliations appears at the end of the paper. ✉email: Helene.Choquet@kp.org

Basal cell carcinoma (BCC) is a keratinocyte carcinoma that is one of the most common malignancies worldwide[1]. Known BCC risk factors include pigmentary traits (e.g., fair skin, light eye color, blonde or red hair), a tendency to sunburn[2], and environmental risk factors, such as UV radiation exposure, ionizing radiation, and immunosuppression[3,4]. BCCs account for approximately 80% of skin cancers, and while skin cancer is far more common in fair-skinned individuals, Hispanic, Asian, and African ancestry individuals account for 4 to 5%, 2 to 4%, and 1 to 2% of skin cancer cases, respectively[5]. Especially, BCCs are the most common skin cancer in European, Hispanic, and Asian ancestry individuals and the second most common in African ancestry individuals (approx. 1.8% of BCCs occur in African ancestry individuals)[6,7].

Keratinocyte carcinoma has a moderate genetic component with a twin or family-based heritability estimate of 43.0%[8] and array-heritability estimates up to 17.0% for BCC[9,10]. Previously published genome-wide association studies (GWAS)[9–14], mainly conducted in European descent populations, have reported more than 78 loci associated with BCC, explaining up to 11.0% of BCC heritability. Despite these discoveries, the BCC genetic landscape remains incomplete, and identification of additional risk loci is needed to further define contributing mechanisms that could be potential therapeutic targets. While genetic and environmental factors that drive the keratinocyte toward BCC and squamous cell carcinoma (SCC) -which is the other type of keratinocyte carcinoma- have been reported[10,15], the genetic overlap between BCC and SCC remains largely uncovered.

Here, we present a large GWA meta-analysis of BCC, including 802,297 individuals of European ancestry (49,905 BCC cases) from the Genetic Epidemiology Research on Adult Health and Aging (GERA) cohort, the Mass-General Brigham (MGB) Biobank cohort, the UK Biobank (UKB) cohort, and the 23andMe, Inc. research cohort (Supplementary Data 1). As a note, the GWAS data set from the 23andMe research cohort consisted of 12,945 BCC cases and 274,252 controls of European ancestry and was previously used in a discovery GWAS[9] and more recently in BCC GWA meta-analyses[10,14]. Similarly,

the UKB data set has been previously included in recent BCC GWA meta-analyses[10,14]. The different steps and data sets used for the current study are summarized in an overview diagram (Fig. 1). We also present a BCC GWA meta-analysis of Hispanic/Latino populations, which consists of 10,468 participants (including 626 BCC cases). We then performed a multi-ancestry BCC GWA analysis to identify additional loci. Next, we conducted a GWA meta-analysis of SCC in 778,893 individuals of European ancestry (16,407 SCC cases) and used the findings to investigate the shared genetic effects between BCC loci and SCC risk. We subsequently fine-mapped our identified associations and examined potential drug compounds targeting BCC risk genes. The associated loci provide relevant pathways underlying BCC susceptibility and highlight candidate genes as potential BCC therapeutic candidates.

## Results

**European ancestry GWA meta-analysis of BCC identifies 116 loci.** In the GWA meta-analysis combining results from four cohorts (i.e., GERA, UKB, MGB, and 23andMe research cohort) and consisting of 49,905 BCC cases and 752,392 controls of European ancestry (Supplementary Data 1), we identified 116 loci associated with BCC ($P < 5\times10^{-8}$; $\lambda = 1.248$ and $\lambda_{1000} = 1.003$, which is reasonable for a sample of this size under the assumption of polygenic inheritance;[16] LDSC intercept = 1.064, 95% CI = 1.038–1.091, which showed no substantial inflation[17]), of which 30 were novel (Table 1, Supplementary Figs. 1-3, and Supplementary Data 2). The effect estimates of the lead SNPs were consistent across the 4 studies, except for SNP rs112108851 at *ZAN* (Table 1 and Supplementary Fig. 4). To identify additional and independent signals, we performed a multi-SNP-based conditional & joint association analysis (COJO)[18], which revealed 67 additional independent SNPs within 28 loci. These include at known BCC loci *ALS2CR12* (chr2 q33.1), near *CLPTM1L* (chr5 p15.33), near *FANCA* (chr16 q24.3), and *TGM3* (chr20 p13) and at the newly locus *FGF7-FAM227B* (chr15 q21.2) identified in the European ancestry GWAS analysis (Supplementary Data 3).

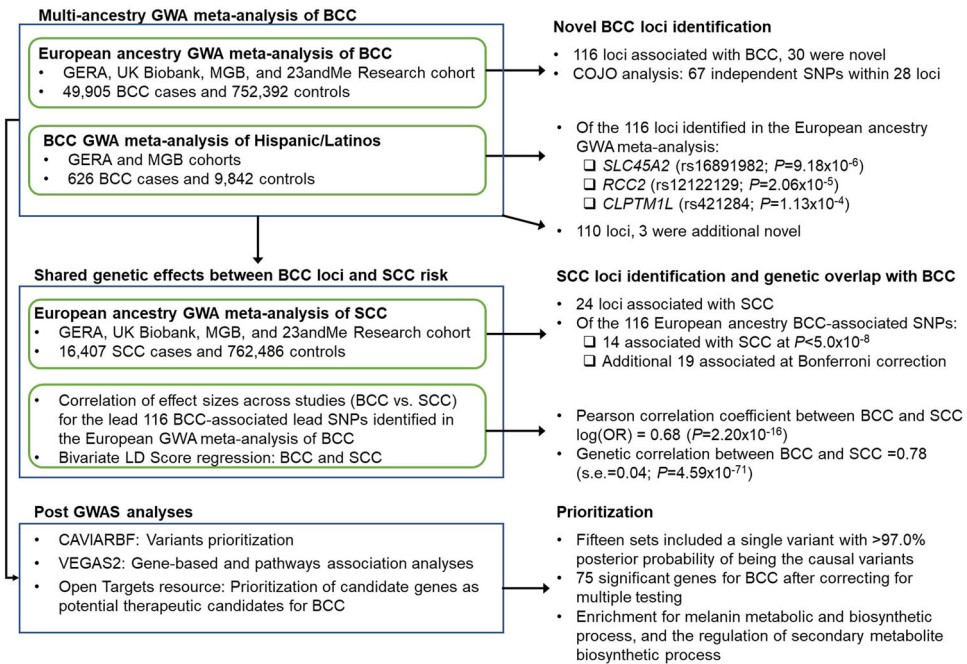

**Fig. 1 Flowchart of the study design.** This figure summarizes the different stages of the study, as well as the data sets and main analyses/results for each stage.

**Table 1 Novel BCC loci identified in the European ancestry, Hispanic/Latino, and/or the multi-ancestry GWA meta-analyses.**

| SNP | CHR | BP | Nearest gene(s) | EA/OA | OR (95%CI) | P | DE | $I^2$ (Q) |
|---|---|---|---|---|---|---|---|---|
| rs56023437 | 1 | 38353403 | INPP5B | T/C | 0.96 (0.94,0.97) | 4.51E-08 | -,-,-,- | 0 (0.7) |
| rs78663305 | 1 | 168497741 | LOC100505918-XCL2 | G/A | 0.94 (0.92,0.96) | 1.77E-08 | -,-,-,- | 0 (0.93) |
| rs6700380 | 1 | 214546861 | PTPN14 | T/C | 0.95 (0.94,0.97) | 2.50E-11 | -,+,-,- | 35.13 (0.2) |
| rs1254197 | 1 | 236361040 | GPR137B | T/G | 0.96 (0.94,0.97) | 2.77E-08 | -,N,-,- | 0 (0.8) |
| rs6726950 | 2 | 10113496 | GRHL1 | G/A | 1.07 (1.04,1.09) | 9.97E-10 | +,+,+,+ | 22.69 (0.27) |
| *rs1364689 | 2 | 34266792 | MYADML-LOC100288911 | A/G | 1.65 (1.38,1.96) | 2.32E-08 | +,+ | 0 (0.63) |
| rs17479393 | 2 | 145653287 | DKFZp686O1327 | T/A | 0.95 (0.93,0.97) | 8.02E-10 | -,-,-,- | 17.07 (0.31) |
| rs10646896 | 2 | 173074802 | DLX2-ITGA6 | CTTT/C | 0.94 (0.92,0.96) | 8.73E-09 | -,N,-,N | 0 (0.84) |
| rs8176526 | 2 | 188345322 | TFPI | T/C | 1.05 (1.03,1.06) | 1.41E-08 | +,+,+,+ | 0 (0.73) |
| rs11684176 | 2 | 198954774 | PLCL1 | T/C | 0.96 (0.95,0.98) | 4.20E-08 | -,-,-,- | 0 (0.98) |
| *rs74269473 | 2 | 217359393 | SMARCAL1-RPL37A | T/C | 2.57 (1.85,3.57) | 2.00E-08 | +,+ | 36.75 (0.21) |
| rs11130229 | 3 | 50011540 | RBM6 | C/T | 0.96 (0.95,0.97) | 9.85E-09 | -,-,-,- | 0 (0.92) |
| rs4580527 | 3 | 101258778 | FAM172BP-TRMT10C | C/A | 0.95 (0.93,0.96) | 6.71E-12 | -,N,-,- | 0 (0.97) |
| rs10936600 | 3 | 169514585 | LRRC34 | T/A | 0.96 (0.94,0.97) | 4.76E-08 | -,- | 0 (0.37) |
| rs13120159 | 4 | 186982959 | SORBS2-TLR3 | C/A | 1.04 (1.03,1.06) | 6.58E-09 | +,+,+,+ | 0 (0.94) |
| rs11743151 | 5 | 38756717 | MIR3650-OSMR | T/C | 0.96 (0.95,0.97) | 3.00E-08 | -,+,-,- | 48.02 (0.12) |
| rs76748680 | 5 | 52378850 | ITGA2 | G/GT | 0.94 (0.93,0.96) | 6.19E-10 | -,N,-,N | 0 (0.85) |
| rs1001114 | 5 | 73802976 | ARHGEF28-ENC1 | G/A | 1.05 (1.03,1.06) | 1.36E-08 | +,+,+,+ | 0 (0.56) |
| rs2964574 | 5 | 151155711 | G3BP1 | A/G | 1.05 (1.03,1.06) | 8.07E-09 | +,+,+,+ | 15.73 (0.31) |
| rs2237159 | 6 | 15388277 | JARID2 | A/G | 1.04 (1.03,1.05) | 3.87E-08 | +,+,+,+ | 0 (0.67) |
| rs29243 | 6 | 29599102 | GABBR1 | A/G | 1.2 (1.13,1.27) | 1.53E-08 | +,+,+,+ | 0 (0.94) |
| rs146968538 | 6 | 74499182 | CD109 | A/AT | 0.95 (0.93,0.96) | 6.50E-09 | -,N,-,N | 55.75 (0.13) |
| rs9388487 | 6 | 126678268 | CENPW-RSPO3 | T/G | 0.95 (0.94,0.97) | 1.23E-08 | -,-,-,N | 0 (0.58) |
| rs112108851 | 7 | 100359454 | ZAN | T/C | 0.95 (0.94,0.97) | 6.05E-09 | -,+,-,- | 46.85 (0.13) |
| rs4871622 | 8 | 126618736 | TRIB1-LOC100130231 | T/C | 0.96 (0.95,0.97) | 4.43E-08 | -,- | 0 (0.49) |
| rs55807015 | 8 | 129003458 | MIR1205-MIR1206 | A/G | 1.05 (1.04,1.07) | 3.44E-11 | +,+,+,+ | 0 (0.78) |
| rs9695995 | 9 | 92454643 | UNQ6494-MIR4290 | G/C | 1.04 (1.03,1.06) | 3.24E-08 | +,+ | 0 (0.32) |
| rs2805831 | 9 | 100466636 | XPA-FOXE1 | A/G | 0.94 (0.92,0.96) | 7.06E-11 | -,-,-,- | 0 (0.57) |
| *rs74603527 | 11 | 99745220 | CNTN5 | G/A | 3.53 (2.34,5.31) | 1.66E-09 | +,+ | 76.99 (0.037) |
| rs9668178 | 12 | 26421434 | SSPN-ITPR2 | A/T | 1.05 (1.03,1.06) | 3.71E-08 | +,+,+,+ | 70.5 (0.02) |
| rs176265 | 12 | 120848516 | MSI1-COX6A1 | G/A | 0.95 (0.93,0.97) | 2.90E-08 | -,-,-,- | 0 (0.82) |
| rs12593917 | 15 | 49836867 | FGF7-FAM227B | T/G | 0.95 (0.94,0.97) | 1.93E-10 | -,-,-,- | 0 (0.73) |
| rs7359174 | 15 | 67413837 | SMAD3 | A/G | 1.07 (1.04,1.09) | 7.49E-09 | +,+,+,+ | 0 (0.74) |
| rs9910411 | 17 | 17609246 | RAI1-SMCR5 | T/G | 1.04 (1.03,1.06) | 1.62E-08 | +,+,+,+ | 22 (0.28) |
| rs61494113 | 19 | 17401859 | ANKLE1-ABHD8 | A/G | 1.05 (1.03,1.07) | 4.25E-08 | +,N,+,+ | 20.64 (0.28) |
| rs12833418 | 23 | 108593972 | IRS4-GUCY2F | T/C | 0.89 (0.86,0.93) | 1.47E-10 | -,N,-,- | 44.43 (0.17) |

*SNP single nucleotide polymorphism, Chr chromosome, Pos position build GRCh37, EA effect allele, OA other allele, OR odds ratio from fixed-effects summary estimates, CI confidence interval, DE direction of effect; direction is listed in order for GERA, MGB, UKB, and 23andMe (for the European ancestry BCC GWA meta-analysis); in order for GERA and MGB (for the Hispanic/Latino ancestry BCC GWA meta-analysis); or in order for European ancestry meta-analysis and Hispanic/Latino ancestry meta-analysis (for the multi-ancestry BCC GWA meta-analysis); N means not included in analysis; $I^2$, heterogeneity index (0-100%); and Q, P-value for Cochrane's Q statistic. Loci marked with a star * were identified in the Hispanic/Latino BCC GWA meta-analysis. We report the nearest gene(s) for each SNP; more complete annotation data (other genes within each locus) can be found in Supplementary Figs. 3 or 7.*

**BCC GWA meta-analysis of Hispanic/Latino populations identifies susceptibility loci.** We then conducted a GWA meta-analysis of BCC in Hispanic/Latinos (626 BCC cases and 9,842 controls) from the GERA and MGB cohorts and we looked-up the 116 loci identified in the European ancestry GWA meta-analysis (Supplementary Data 4). The GWA meta-analysis of BCC in Hispanic/Latinos resulted in the identification of 4 genome-wide significant loci, of which 3 have not been previously reported to be associated with BCC in European ancestry populations: near MYADML (chr2 p22.3), PEBP4 (chr8 p21.3), and CNTN5 (chr11 q22.1) (Table 1, Supplementary Figs. 5-7 and Supplementary Data 5). We also found that three of the lead SNPs identified in the European ancestry GWA meta-analysis were associated with BCC at Bonferroni significance. These include: a missense variant rs16891982 (p.Leu374Phe) at SLC45A2 (OR = 1.43; $P = 9.18 \times 10^{-6}$), an upstream variant rs12122129 at RCC2 (OR = 1.34; $P = 2.06 \times 10^{-5}$), and an intronic variant rs421284 at CLPTM1L (OR = 0.77; $P = 1.13 \times 10^{-4}$) (Supplementary Data 4 and Supplementary Fig. 7).

**European-derived PRS predicted BCC risk in Hispanic/Latinos.** We first constructed two polygenic risk scores (PRSs) for BCC based on the GWAS summary statistics from the European ancestry GWA meta-analysis using two different P-value thresholds (i.e., $P < 5.0 \times 10^{-8}$ and $P < 1.0 \times 10^{-6}$) (see Methods). Each predictive model included a PRS, along with age and sex, and was poorly predictive of BCC risk in the Hispanic/Latino sample from the GERA cohort with an area under the curve (AUC) value of 0.545 and 0.573, depending on the threshold of significance for the SNPs (P-value $< 5.0 \times 10^{-8}$ and $P < 1.0 \times 10^{-6}$, respectively) (Supplementary Fig. 8).

**Multi-ancestry GWA meta-analysis identifies 3 additional novel BCC loci.** We further conducted a multi-ancestry GWA meta-analysis by combining results from the European ancestry GWA meta-analysis and the Hispanic/Latino GWA meta-analysis (totaling 50,531 BCC cases and 762,234 controls). We identified 110 loci, of which 3 were not identified in the European ancestry GWA meta-analysis and were not previously reported to be associated with BCC risk (Fig. 2, Supplementary Fig. 9, and Supplementary Data 6). These include: LRRC34, TRIB1-LOC100130231, and UNQ6494-MIR4290. Regional plots of the association signals at the 3 novel loci are presented in Supplementary Fig. 10. We found a high concordance of effect sizes

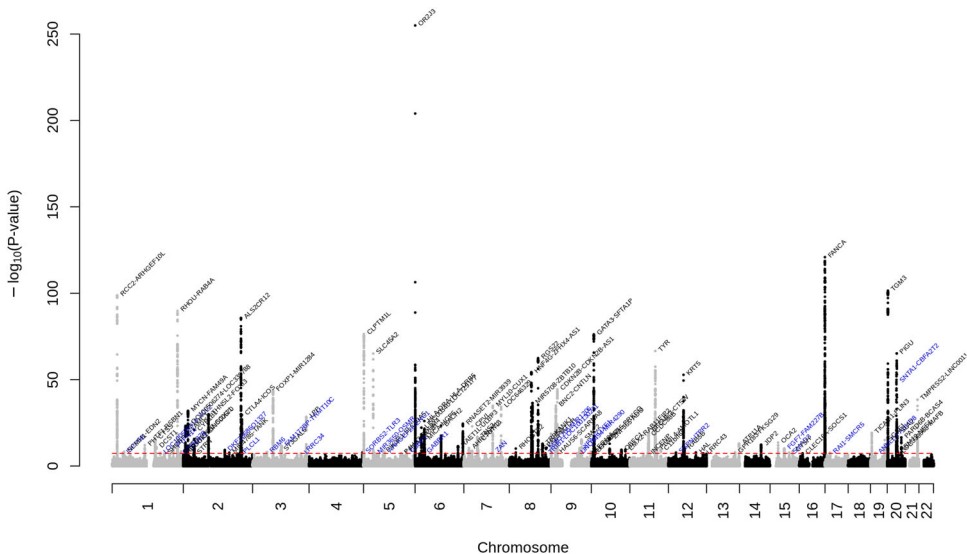

**Fig. 2 Manhattan plot of the multi-ancestry GWA meta-analysis of BCC.** The y-axis represents the -log₁₀(P-value); all P-values derived from logistic regression model are two-sided. The red dotted line represents the threshold of $P = 5 \times 10^{-8}$ which is the commonly accepted threshold of adjustments for multiple comparisons in GWAS. Locus names in blue are for the novel loci and the ones in dark are for the previously reported ones.

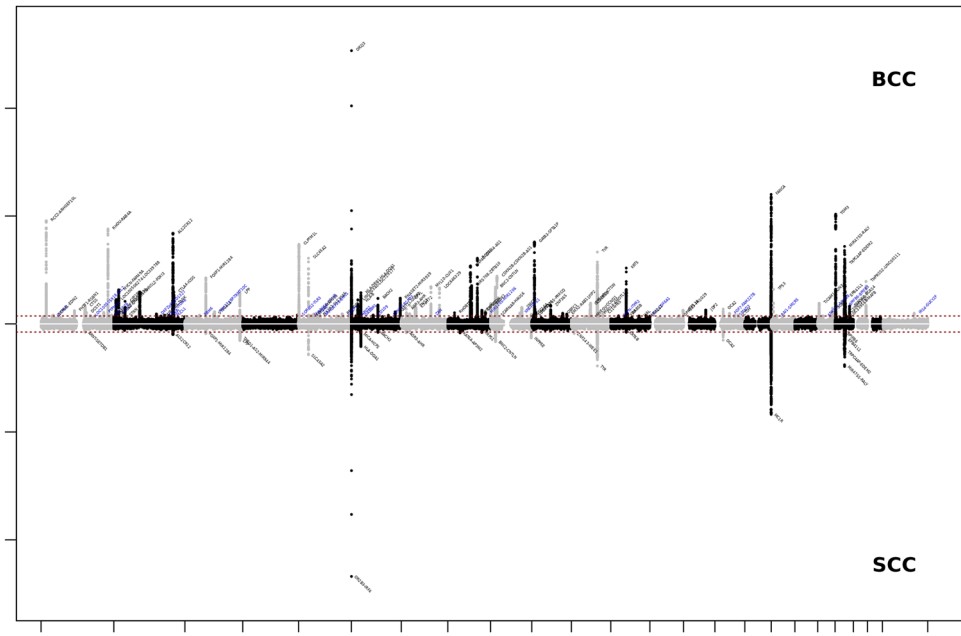

**Fig. 3 Chicago plot of the European ancestry GWA meta-analyses of KC.** Results from the European ancestry GWA meta-analysis of BCC are presented on upper panel, while results from the European ancestry GWA meta-analysis of SCC are presented on the lower panel. Both European ancestry GWA meta-analyses combined results from GERA, MGB, UKB, and 23andMe cohorts. The y axis represents the -log10(P-value); all P-values derived from logistic regression model are two-sided. The red dotted line represents the threshold of $P = 5 \times 10^{-8}$ which is the commonly accepted threshold of adjustments for multiple comparisons in GWAS. Locus names in blue are for the novel loci and the ones in dark are for the previously reported ones.

between the 2 ancestry meta-analyses (Pearson's coefficient correlation=0.75, $P = 3.38 \times 10^{-19}$) (Supplementary Fig. 11a, b).

**Shared genetic effects between BCC loci and SCC risk**. We then conducted a European ancestry GWA meta-analysis of SCC by combining data from GERA, UKB, MGB, and 23andMe research cohort (totaling 16,407 SCC cases and 762,486 controls) (Supplementary Data 1). Our genome-wide analysis of SCC revealed 24 genome-wide significant loci ($\lambda = 1.104$ and $\lambda_{1000} = 1.003$, which is reasonable for a sample of this size under the assumption of polygenic inheritance[16]; LDSC intercept = 1.056, 95% CI =

1.032 − 1.080, which showed no substantial inflation[17]), (Supplementary Figs. 12-13 and Supplementary Data 7). Of the 116 genome-wide-significant BCC SNPs identified in the European ancestry GWA meta-analysis, 33 were significantly associated with SCC after Bonferroni correction ($P < 0.05/116 = 4.31 \times 10^{-4}$), including 14 at genome-wide level of significance (Fig. 3 and Supplementary Data 8). The Pearson correlation coefficient between BCC effect size (log(OR)) and the SCC log(OR) was 0.68 ($P = 2.20 \times 10^{-16}$; Fig. 4). Using bivariate LD Score regression, we estimated the genome-wide genetic correlation between BCC and SCC to be 0.78 (s.e.=0.04; $P = 4.59 \times 10^{-71}$). We also conducted genome-wide genetic correlation analyses between the 2 non-

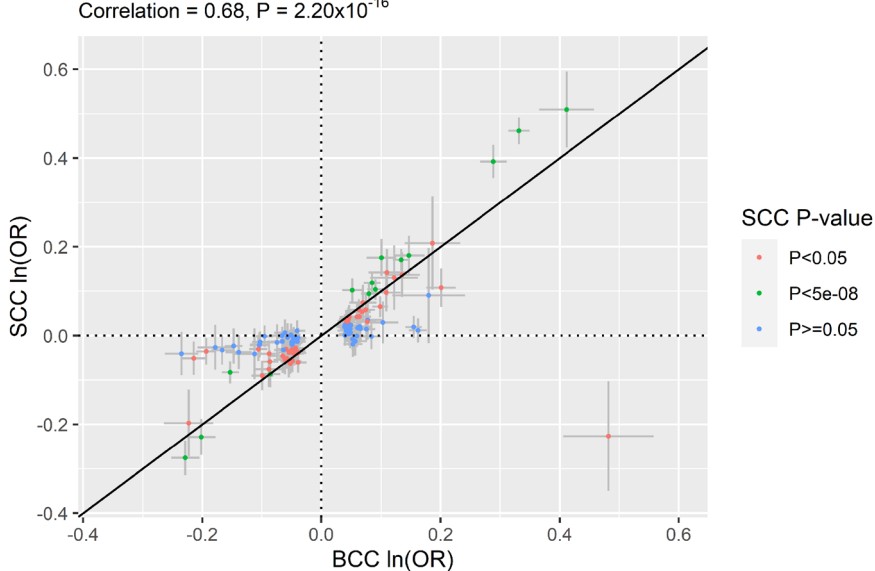

**Fig. 4 Correlation of effect sizes across studies (BCC vs. SCC) for the lead 116 BCC-associated SNPs identified in the European GWA meta-analysis of BCC.** Comparison of the direct effect sizes (log(OR)) for BCC in 49,905 BCC cases and 752,392 controls of European ancestry (x-axis) (Supplementary Data 2) and for SCC in 16,407 SCC cases and 762,486 controls of European ancestry (y-axis) (Supplementary Data 8). The 116 BCC-associated SNPs are shown as dots; and the error bars correspond to the 95% CIs and are displayed in gray. While SNPs in green show $P < 5.0 \times 10^{-8}$ with SCC and those in red show $P < 0.05$, those in blue are not associated with SCC ($P \geq 0.05$).

melanoma skin cancers (BCC and SCC) and by cohort using LD score regression[17]. We found a high degree of genetic correlations between BCC and SCC and across the cohorts (Supplementary Fig. 14 and Supplementary Data 9). For instance, BCC GWAS from GERA was highly correlated with SCC GWAS from 23andMe ($r_g = 0.88$; se = 0.13; $P$-value = $5.58 \times 10^{-11}$), while the BCC GWAS from UKB was more moderately correlated with the SCC GWAS from GERA ($r_g = 0.59$; se = 0.09; $P$-value = $3.71 \times 10^{-11}$).

**Variants prioritization.** We used a Bayesian approach (CAVIARBF)[19] to compute variant likelihoods and to explain the observed association at each BCC-associated locus and derive the smallest set of variants with a 95% probability of including the causal origin of the signals (95% credible set). Fifteen sets included a single variant (Supplementary Data 10), including rs1800440 (*CYP1B1*), rs16891982 (*SLC45A2*), rs62389423 (*OR2J3*), rs9266184 (*HLA-B*), rs117132860 (*AGR3-AHR*), rs117744081 (*CPVL*), rs2241261 (*RHOBTB2*), rs1126809 (*TYR*), rs11170164 (*KRT5*), rs1800407 (*OCA2*), rs12931267 (*FANCA*), rs78378222 (*TP53*), rs7508601 (*SCAF1*), rs62209647 (*CHMP4B-MIR4755*), and rs6059655 (*MIR4755-RALY*) with >97.0% posterior probability of being the causal variants. This suggests that those single variants may be the causal origin of the associations observed in their respective BCC-associated loci.

**Gene-based and pathways association analyses.** To prioritize genes within the identified BCC loci, we performed a gene-based association analysis using the Versatile Gene-based Association Study (VEGAS2v02) integrative tool[20]. We used the multi-ancestry GWA meta-analysis results as input for this gene-based association analysis. We identified 75 significant genes for BCC after correcting for multiple testing (Bonferroni correction was set as $P < 2.28 \times 10^{-6}$ (0.05/21,961 genes tested)), with the strongest association for the *SLC45A2* gene ($P = 5.59 \times 10^{-11}$) followed by *RCC2* ($P = 1.28 \times 10^{-9}$), and *CLPTM1L* ($P = 3.50 \times 10^{-9}$) (Supplementary Data 11). We then conducted a pathway analysis using VEGAS2v02 and based on the gene-based association

results to assess enrichment in 9732 pathways (or gene-sets) derived from the Biosystem's database. We found that pathways involving the melanin metabolic and biosynthetic process as well as the regulation of secondary metabolite biosynthetic process were significantly enriched after correcting for multiple testing (Bonferroni correction was set as $P < 5.14 \times 10^{-6}$ (0.05/9732 pathways tested)) (Supplementary Data 12).

**Prioritization of candidate genes as potential drug targets for BCC.** By leveraging multi-omics datasets (i.e., eQTL, chromatin interaction) using Open Targets[21], we prioritized 14 candidate genes that could be potential drug targets for BCC (Supplementary Data 13). These include *AHR, CCND1, CTLA4, CTSS, FGF1, GABBR1, GLRA1, HLA-DRB5, ICOS, IL2RA, PIK3R1, TLR3, TP53,* and *VDR.* We discuss the relevance of some of these candidate genes in the discussion section below.

## Discussion

In this study, by leveraging GWAS data from four cohorts, we identified 122 risk loci for BCC, of which 36 were not previously identified. We also reported an association of the well-known pigment gene *SLC45A2* as well as associations at *RCC2* and *CLPTM1L* with BCC in Hispanic/Latinos. We observed significant shared genetic effects between BCC loci and SCC risk. Finally, we prioritized causal variants, candidate genes that could be potentially relevant drug targets for BCC, and pathways, using bioinformatic functional analyses.

Among the novel SNPs associated with BCC susceptibility in our European ancestry meta-analysis, we identified rs6700380 which lies in an intron of *PTPN14* (1q41). *PTPN14* encodes the protein tyrosine phosphatase non-receptor type 14, which is a tumor-suppressor gene, and loss-of-function and deleterious missense mutations in this gene have been linked to BCC tumorigenesis[22,23]. We also identified an association between rs6726950 (which is an intronic variant of *GRHL1*, 2p25.1) with BCC. *GRHL1* is a member of the grainyhead family of transcription factors that has an important role in maintaining the skin barrier[24,25], and has been implicated in trichogerminoma[26],

a subset of follicular tumors characterized by immature features and numerous Merkel cells. We also identified an association between an intronic variant rs76748680 of *ITGA2* (2q37.1) with BCC. *ITGA2* encodes the alpha subunit of a transmembrane receptor for collagens and related proteins and has been involved in the regulation of the PI3K/AKT signaling[27], an important pathway implicated in the Cowden syndrome, which is an inherited disorder associated with an increased risk of keratinocyte carcinoma[28]. Interestingly, the VEGAS2 gene-based association analysis also prioritized those genes ($P = 2.13 \times 10^{-4}$ for *PTPN14*; $P = 1.04 \times 10^{-5}$ for *GRHL1*; and $P = 3.45 \times 10^{-5}$ for *ITGA2*), suggesting that they could be the potential candidate genes underlying those associations. Thus, these identified loci provide novel insight and additional evidence about the potential genes and pathways/systems underlying BCC susceptibility.

Our study also suggested, for the first time to our knowledge, genetic risk factors for BCC in Hispanic/Latinos. These include, notably, *SLC45A2*, *RCC2*, and *CLPTM1L*. *SLC45A2* is a pigmentation-related gene that encodes a transporter protein that mediates melanin synthesis and has been previously reported as a BCC risk locus in European descent populations[13]. Mutations in *SLC45A2* gene are a cause of a form of albinism, and polymorphisms in this gene are associated with variations in pigmentation traits (skin and hair color) and tanning ability[29–33]. *RCC2* encodes the regulator of chromosome condensation 2 and has been previously reported as a candidate gene and a susceptibility locus for BCC risk[9,11]. RCC2 has been involved in the etiology of various cancers and has been shown to be a biomarker for colorectal cancer[34–38]. *CLPTM1L* encodes a membrane protein whose overexpression in cisplatin-sensitive cells causes apoptosis. Polymorphisms at this locus have been reported to increase susceptibility to many cancer types, including pancreatic, lung, bladder cancer, and melanoma[39,40]. Our findings were also supported by our VEGAS2 gene-based analysis which also highlighted *SLC45A2*, *RCC2*, and *CLPTM1L* as BCC-associated genes. We also identified 4 loci (near *MYADML* (chr2 p22.3), *SMARCAL1-RPL37A* (chr2 q35), *PEBP4* (chr8 p21.3), and *CNTN5* (chr11 q22.1)) associated with BCC at a genome-wide level of significance in the Hispanic/Latino GWA meta-analysis. Because of the limited sample size of this analysis, these loci will need to be validated in future studies of Hispanic/Latino individuals to confirm their role in BCC susceptibility.

Our results also prioritized 14 candidate genes within the identified BCC risk loci that are targeted by existing drugs, some of those are already in use/clinical trials for various cancer types, inflammatory skin diseases, or systemic diseases. For instance, VDR is targeted by calcitriol, a Vitamin D receptor agonist that is currently used to treat various diseases such as cancers, psoriasis, and atopic eczema[41,42]. Interestingly, a Mendelian randomization study demonstrated a causal effect of vitamin D on BCC susceptibility, such as higher 25-hydroxyvitamin D levels increase risk of BCC[43]. Consistently, clinical trials demonstrated a beneficial role of vitamin D compounds in the treatment of actinic keratosis and other clinical trials are in progress to evaluate calcitriol therapy for BCC[44]. Some other drug candidates targeting proteins encoded by BCC-associated loci are also currently under consideration in ongoing clinical trials[45–47] for treating various cancer types and Cowden syndrome, including samotolisib, a PI3-kinase class I inhibitor targeting PIK3R1. Also, ipilimumab, a cytotoxic T-lymphocyte protein 4 inhibitor, targets CTLA4, and aldesleukin, an interleukin-2 receptor agonist, targets IL2RA; current clinical trials are testing therapies based on these drugs for Merkel cell skin cancer and other skin cancers[48–50]. Our findings are consistent with a recent genetic study that also prioritized *CTLA4* as a potential drug target for BCC treatment[14]. Moreover, given that our pathway analyses highlighted the involvement of the melanin metabolic and biosynthetic process in BCC pathogenesis, drugs targeting genes (i.e., *TRPC1*, *SLC45A2*, *TYRP1*, *TYR*, *PMEL*, *DCT*, *OCA2*, *MYO5A*, *MC1R*, *CTNS*, *ASIP*, *DDT*, and *BCL2*) involved in those pathways could be potential therapies for BCC. Most of these genes are pigmentation-related genes (i.e., *SLC45A2*, *TYR*, *OCA2*, *MC1R*, and *ASIP*) and have a biological importance of lighter pigmentation in susceptibility to keratinocyte carcinoma. Even if agonists for some of these genes exist (e.g., Afamelanotide, a α-melanocyte-stimulating hormone (MSH) analogue that binds *MC1R*) to treat some skin diseases[51], because of the importance of sun exposure in keratinocyte carcinoma[52], the modification of the melanin biosynthesis by those drugs as a treatment for BCC is questionable. Thus, existing preventative interventions, such as sun avoidance, remain presumably the most appropriate approach to decrease keratinocyte carcinoma risk in those cases. Moreover, further investigations are required to confirm the functionality of these genes in vivo and in vitro which may support the suitability of these potential new druggable genes as alternative treatments for BCC.

We recognize some potential limitations of our study. First, it is important to note phenotypic differences for BCC between the 4 study cohorts. While in the GERA cohort, BCC cases were identified from electronic pathology records using a validated SNOMED code-based algorithm, in the MGB and UK Biobank, BCC cases were identified based on International Classification of Disease, Ninth (ICD-9) and/or Tenth (ICD-10) diagnosis codes, and in 23andMe research cohort, participants self-reported a history of BCC. This could have led to misclassification of BCC phenotype. However, our meta-analysis combining GERA, MGB, UKB, and 23andMe results showed consistency of the SNPs effect estimates between cohorts, except for rs112108851 at *ZAN* for which we observed an inconsistent direction of effects in MGB compared to the other cohorts. Further, we demonstrated substantial genetic correlation between BCC phenotypes across the different cohorts. Second, in order to maximize power to detect novel BCC and SCC risk loci, we did not separate our sample into a separate discovery and replication set. However, our results replicated most of previous known loci, especially from a recent large multi-trait genetic analysis that reported 78 risk loci for BCC[14]. Our study also has important strengths, including the large sample size of up to 812,765 participants (50,531 BCC cases) from 4 different research studies. Furthermore, to the best of our knowledge, this is the first study that enabled cross populations comparison (i.e., between non-Hispanic whites and Hispanic/Latinos) for BCC genetic risk factors.

In conclusion, our large multi-ancestry GWA meta-analysis findings provide new insights into the genetic basis of BCC and SCC susceptibility and expand the list of shared genetic risk factors between BCC and SCC. Our study findings also uncover *SLC45A2*, *RCC2*, and *CLPTM1L* as promising genetic risk factors for BCC in Hispanic/Latinos. Finally, our fine-mapping and bioinformatic annotation analyses provide functional relevance for candidate genes, involved in the pathogenesis of BCC, that could be potentially targeted for treating BCC.

## Methods

**European ancestry GWA meta-analysis of BCC.** The GWA meta-analysis of BCC consisted of four cohorts (Supplementary Data 1). The GWAS data set from the Kaiser Permanente Norther California Genetic Epidemiology Research in Adult Health and Aging (GERA) cohort encompassed 20,529 BCC cases and 58,051 controls of European ancestry. The GWAS data set from the Mass-General Brigham (MGB) Biobank cohort encompassed 3,146 BCC cases and 31,196 controls of European ancestry. The GWAS data set from the UK Biobank cohort consisted of 13,285

BCC cases and 388,893 controls of European ancestry. The GWAS data set from the 23andMe research cohort consisted of 12,945 BCC cases and 274,252 controls of European ancestry and was used in a previously reported GWAS[9]. In total, 49,905 BCC cases and 752,392 controls of European ancestry were included in this first GWA meta-analysis of BCC. Samples for each cohort (GERA, MGB, UK Biobank, and 23andMe Research cohort) were collected with informed consent. Full details on genotyping, imputation, and quality control for each cohort are reported in Supplementary Note 2.

**BCC GWA meta-analysis of Hispanic/Latino populations**. The GWA meta-analysis of BCC in Hispanic/Latinos consisted of two cohorts (Supplementary Data 1), GERA and MGB, totaling 626 BCC cases and 9842 controls.

**Multi-ancestry GWA meta-analysis of BCC**. A multi-ancestry GWA meta-analysis combining results from the above-mentioned European ancestry GWA meta-analysis, and the Hispanic/Latino GWA meta-analysis consisted of 50,531 BCC cases and 762,234 controls.

**European ancestry GWA meta-analysis of SCC**. The GWA meta-analysis of cutaneous SCC consisted of four cohorts (Supplementary Data 1). The GWAS data set from the GERA cohort encompassed 7687 SCC cases and 60,068 controls of European ancestry and was used in a previously reported GWAS[53]. The GWAS data set from the Mass-General Brigham (MGB) Biobank cohort encompassed 627 SCC cases and 32,967 controls of European ancestry. The GWAS data set from the UK Biobank cohort consisted of 1514 SCC cases and 388,893 controls of European ancestry. The GWAS data set from the 23andMe research cohort consisted of 6579 SCC cases and 280,558 controls of European ancestry and was used in a previously reported GWAS[54]. In total, 17,181 SCC cases and 713,994 controls of European ancestry were included in this first GWA meta-analysis of SCC.

**BCC and SCC phenotyping**. In the **GERA** cohort, BCC cases were identified from electronic pathology records. After excluding individuals with any evidence of metastatic BCCs (SNOMED codes M80906, M809061, M809063, M80909, M809092, M809093, M80946, M809492, M809493, M80960), our control group included all the non-cases after excluding individuals who had a current or prior cancer registry history of cancer, or benign or in-situ tumors, or had a self-reported cancer at the time of enrollment, as previously done[55]. Further, cSCC cases were defined as subjects whose pathology records were consistent with incident cSCC (invasive or in situ, excluding anogenital and mucosal SCCs); controls were subjects with no pathology records consistent with cSCC[53], and similar to our BCC control group, we excluded individuals who had a current or prior cancer registry history of cancer, or benign or in-situ tumors, or had a self-reported cancer at the time of enrollment. In **MGB**, BCC cases were defined as those with BCC diagnosis using ICD-10 (C44.01-C44.91) codes, while those without BCC diagnosis were considered controls. Further, cSCC cases were defined as those with cSCC diagnoses using International Classification of Disease (ICD), Ninth or Tenth Clinical Modification (CM) codes (ICD-9-CM: 173.0-173.9; ICD-10-CM: C44.0-C44.9) and were subsequently validated by electronic pathology reports review. Those without cSCC diagnosis were considered controls. In **UKB**, BCC or SCC cases were defined as participants with an ICD-9 or ICD-10 diagnosis code for BCC or SCC and based on histology data (e.g., field ID: 40011) (see full process in Supplementary Notes 1 and 2). As done before[55], we excluded UKB participants who self-

reported cancer at the time of enrollment and/or who had a current or prior cancer registry history of cancer, or benign or in-situ tumors, from the control groups. In the **23andMe research cohort**, participants who self-reported a history of BCC or SCC cases were assigned to the BCC or SCC cases[9,54].

**GWAS and adjustment**. In **GERA**, we first analyzed each ethnic group (non-Hispanic white, and Hispanic/Latino) separately. We ran a logistic regression of BCC and SCC and each SNP using a recent approach accounting for relatedness that fits a whole-genome regression model, implemented in REGENIEv2.0.2[56]. GWAS analyses were adjusted for age, sex, and ancestry principal components (PCs), which were previously[57] assessed within each ethnic group using Eigenstrat[58] v4.2. We included as covariates the top ten ancestry PCs and the percentage of Ashkenazi (ASHK) ancestry for the non-Hispanic whites, whereas we included the top six ancestry PCs for the Hispanic/Latino ethnic group[57]. In **MGB**, PLINK v1.90 was also used to conduct the genome-wide association analysis, adjusted for age, sex, and the top ten PCs. All phenotyping analyses were conducted using R[59] (version 3.6.2, http://www.R-project.org/) and STATA 15.0 (StataCorp. 2017. Stata Statistical Software: Release 15. College Station, TX: StataCorp LLC). In **UKB**, REGENIEv2.0.2[56] was also used to conduct the GWA analyses, and age, sex, and genetic ancestry PCs were included as covariates and the analyses presented in this paper were carried out under UK Biobank Resource project #14105. In the **23andMe** GWA analyses, age, sex and five PCs were included as covariates[9,54]. Analyses were restricted to SNPs with minor allele frequency (MAF) > 1% and an imputation quality score >0.5. Further, all analyses were conducted assuming an additive genetic model, and variants with $P < 5.0 \times 10^{-8}$ were considered genome-wide significant.

**GWA meta-analyses**. Fixed-effects meta-analyses (of BCC and SCC) were conducted to combine the results from GERA, MGB, UKB, and 23andMe research cohort using the inverse variance-based method, as implemented in PLINK. We also estimated heterogeneity index, $I^2$ (0–100%) and P-value for Cochrane's Q statistic among studies. For each locus, we defined the top SNP as the most significant variant within a 2 Mb window. Novel loci were defined as those that were located over 1 Mb apart from any previously reported locus[9,10,14,15,53,54,60,61], except for the genomic region 20q11 (around *ASIP*) where strong and long-range linkage disequilibrium (LD) has been observed and for which an extended window of 2.5 Mb was used. Loci were also defined as novel if the identified lead SNPs were not in linkage disequilibrium (LD) with previously reported SNPs using LDlink tool[62]. Locus names reported in the manuscript correspond to the nearest gene(s) for each lead SNP based on the National Center for Biotechnology Information dbSNP tool (www.ncbi.nlm.nih.gov/snp/).

**Conditional & joint (COJO) analysis**. A multi-SNP-based conditional & joint association analysis (COJO)[18] was performed on the combined European ancestry GWA meta-analysis results to potentially identify additional and independent signals. To calculate LD patterns, we used 10,000 randomly selected samples from GERA non-Hispanic white ethnic group as a reference panel. A P-value less than $5.0 \times 10^{-8}$ was considered significant.

**Genetic correlation between BCC and SCC**. A LD score regression[17,63] was conducted, using the LDSC v1.0.1 command line tool (https://github.com/bulik/ldsc), to estimate the genome-wide genetic correlations between BCC and SCC across cohorts. GWAS summary statistics of the European ancestry samples from

the GERA, MGB, UKB, and 23andMe research cohorts for BCC and SCC were used as input data.

**Polygenic risk scores for BCC and prediction models.** We constructed two PRSs for BCC based on the GWAS summary statistics from the European ancestry GWA meta-analysis, using two different P-value thresholds (i.e., $P < 5.0 \times 10^{-8}$ and $P < 1.0 \times 10^{-6}$). LD clumping was performed using a 10000 kb LD window and a $r^2$ cutoff of 0.005. The first PRS for BCC (PRS1) included 136 clumped independent SNPs at $P < 5.0 \times 10^{-8}$; and the second PRS (PRS2) included 194 clumped independent SNPs at $P < 1.0 \times 10^{-6}$. The two PRSs were calculated as a weighted sum of risk alleles by their estimated effect sizes[64]. To assess the potential value of those two European-derived PRSs to predict BCC risk, regression-based models were tested in the Hispanic/Latino sample from the GERA cohort, which was not part of any of the analyses through which the genetic associations were identified. Each model included PRS (PRS1 or PRS2), age, and sex. The AUC were calculated, and the receiver operating characteristic curves were drawn. The R programming language and software environment for statistical computing was used for calculating the two PRSs for BCC, as well as for conducting the logistic regression models ('glm') and for calculating AUC and generating the receiver operating characteristic curves ('pROC').

**Variants prioritization.** To prioritize variants within the 116 BBC loci identified in the European ancestry GWA meta-analysis, we used a Bayesian approach (CAVIARBF)[19], which is available publicly at https://bitbucket.org/Wenan/caviarbf. Each variant's capacity to explain the identified signal within a 2 Mb window ($\pm 1.0$ Mb with respect to the original top variant) was computed for each identified genomic region. Then, the smallest set of variants that included the causal variant with 95% probability (95% credible set) was derived.

**Genes and pathways prioritization.** To prioritize genes and biological pathways, we conducted gene-based and pathways analyses using the Versatile Gene-based Association Study - 2 version 2 (VEGAS2v02) web platform[20]. We first performed a gene-based association analysis on the BCC multi-ancestry GWA meta-analysis results using the default '-top 100' test that uses all (100%) variants assigned to a gene to compute gene-based P-value. Gene-based analyses were conducted on each of the individual ethnic groups (European and Hispanic/Latino) using the appropriate reference panel: 1000 Genomes phase 3 European population, and 1000 Genomes phase 3 American population, respectively. We then meta-analyzed the 2 ethnic groups gene-based results using Fisher's method for combining the P-values. As 22,673 genes were tested, the P-value adjusted for Bonferroni correction was set as $P < 2.28 \times 10^{-6}$ (0.05/22,961).

Second, we performed pathways analyses based on VEGAS2 gene-based P-values. We tested enrichment of the genes defined by VEGAS2 in 9732 pathways or gene-sets (with 17,701 unique genes) derived from the Biosystem's database (https://vegas2.qimrberghofer.edu.au/biosystems20160324.vegas2pathSYM). We adopted the resampling approach to perform pathway analyses using VEGAS2 derived gene-based P-values considering the default '-10 kbloc' parameter as used before[65]. We then meta-analyzed the 2 ethnic groups gene-based results using Fisher's method for combining the P-values. As 9732 pathways or gene-sets from the Biosystem's database were tested, the P-value adjusted for Bonferroni correction was set as $P < 5.14 \times 10^{-6}$ (0.05/9732).

**Prioritization of drug targets.** We used Open Targets[21] (http://genetics.opentargets.org), to search for drugs currently in use or in clinical trials for treating other skin cancer or systemic diseases that target the BCC risk genes identified in the current study. Potential druggable genes were prioritized if they were close to the most significant variants. These drugs can be potentially repurposed as alternative treatments for BCC, owing to in vivo and in vitro confirmation of the functionality of the target genes in the pathogenesis of BCC.

**Reporting summary.** Further information on research design is available in the Nature Portfolio Reporting Summary linked to this article.

## Data availability

To protect individual's privacy, complete GERA data are available upon approved applications to the KP Research Bank Portal (https://researchbank.kaiserpermanente.org/). A subset of the GERA cohort consented for public use can be found at NIH/dbGaP: phs000674.v3.p3. The GERA GWAS data are available from the corresponding author on reasonable request. The MGB GWAS data are available by request with required approval from the Mass General Brigham Institutional Review board from https://www.massgeneralbrigham.org/en/research-and-innovation/for-researchers-and-collaborators. UK Biobank data, including BCC and SCC GWASs are available by request through the UK Biobank Access Management System https://www.ukbiobank.ac.uk/. The BCC and SCC GWAS results[9,54] from 23andMe are available by request from https://www.23andme.com/. Restrictions apply to the availability of these data (please see https://research.23andme.com/dataset-access/), which were used under license for this study, and are not publicly available. Pathways or gene-sets were derived from the Biosystem's database which can be accessed through the following link (https://vegas2.qimrberghofer.edu.au/biosystems20160324.vegas2pathSYM).

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

## Acknowledgements

We are grateful to the Kaiser Permanente Northern California members who have generously agreed to participate in the Kaiser Permanente Research Program on Genes, Environment, and Health. Support for participant enrollment, survey completion, and biospecimen collection for the RPGEH was provided by the Robert Wood Johnson Foundation, the Wayne and Gladys Valley Foundation, the Ellison Medical Foundation, and Kaiser Permanente Community Benefit Programs. Genotyping of the GERA cohort was funded by a grant from the National Institute on Aging, the National Institute of Mental Health, and the National Institute of Health Common Fund (RC2 AG036607). The current study was funded by a grant from the National Cancer Institute (NCI) R01CA2416323 (HC and MMA). We would like to acknowledge Victor Herrera for putting together the Supplementary Data and Supplementary Information.

## Author contributions

H.C. and M.M.A. contributed to study conception and design. T.J.H. and E.J. were involved in the genotyping and quality control of the GERA samples. T.J.H. performed the imputation analyses in the GERA cohort. C.J. and J.Y. performed GWAS analyses in GERA and UKB and meta-analyses as well as post-GWAS analyses. Y.K. and M.M.A.

extracted phenotype and other data for the M.G.B. participants and Y.K. conducted GWA analysis in M.G.B. 23andMe Research Team performed GWAS analyses in the 23andMe Research cohort. H.C. and M.M.A. interpreted the results of analyses. H.C. drafted the manuscript. H.C. and M.M.A. critically revised the manuscript for key intellectual content.

## Competing interests

The authors declare no competing interests. H.C. is an Editorial Board Member for *Communications Biology*, but was not involved in the editorial review of, nor the decision to publish this article.

## Additional information

## 23andMe Research Team

Stella Aslibekyan[7], Adam Auton[7], Elizabeth Babalola[7], Robert K. Bell[7], Jessica Bielenberg[7], Katarzyna Bryc[7], Emily Bullis[7], Daniella Coker[7], Gabriel Cuellar Partida[7], Devika Dhamija[7], Sayantan Das[7], Sarah L. Elson[7], Teresa Filshtein[7], Kipper Fletez-Brant[7], Pierre Fontanillas[7], Will Freyman[7], Pooja M. Gandhi[7], Karl Heilbron[7], Barry Hicks[7], David A. Hinds[7], Ethan M. Jewett[7], Yunxuan Jiang[7], Katelyn Kukar[7], Keng-Han Lin[7], Maya Lowe[7], Jey McCreight[7], Matthew H. McIntyre[7], Steven J. Micheletti[7], Meghan E. Moreno[7], Joanna L. Mountain[7], Priyanka Nandakumar[7], Elizabeth S. Noblin[7], Jared O'Connell[7], Aaron A. Petrakovitz[7], G. David Poznik[7], Morgan Schumacher[7], Anjali J. Shastri[7], Janie F. Shelton[7], Jingchunzi Shi[7], Suyash Shringarpure[7], Vinh Tran[7], Joyce Y. Tung[7], Xin Wang[7], Wei Wang[7], Catherine H. Weldon[7], Peter Wilton[7], Alejandro Hernandez[7], Corinna Wong[7] & Christophe Toukam Tchakouté[7]

[7]23andMe Inc, Sunnyvale, CA, USA.

