## [Peer Review File · Communications Biology]

Reviewers' comments:

Reviewer #1 (Remarks to the Author):

The authors report on a large GWAS meta-analysis of BCC and SCC. Importantly they have extended their analysis to include individuals with Hispanic/Latino ancestry.

Major comments

1. Introduction line 40 and/or 43: Please report the twin or family-based heritability estimates for BCC and SCC to contextualise the SNP-h² estimates e.g. Mucci 2016 PMID: 26746459

2. Intro line 41: A recent multi-trait GWAS of BCC and SCC is not cited PMID: 36496446. The overlap in terms of findings with that work and this one should be discussed.

3. Intro lines 49-51. Please indicate which GWAS have been previously reported or are new and the proportion of new samples relative to previous publications.

4. Results line 66 - some of the novel loci are reported in (or are in LD with previously reported loci) in Adolphe 2021 PMID: 33549134, Liyanage 2019 PMID 31174203, Seviiri 2022 PMID 36496446 etc. Methods line 318-19 suggests only distance was used to identify new loci. This must also include an appropriate, strong, filter for LD as well e.g. $r^2 < 0.05$ and/or an approach like GCTA cojo, as used later in the methods, to determine which loci are independent of other lead SNPs. The aim of reporting novel loci should be to identify new biology/genes/pathways reliably associated with the risk of SCC or BCC, and SNPs whose association is driven by previously identified regions aren't achieving this goal. While I have highlighted specific examples below, both LD and distance to prior reports should be considered for all findings:

a. BCC: chr6/rs29243. LD is complex and far-reaching in the HLA region. Do the cojo results or variant prioritisation (sup table 8) support this variant as being independent of other known BCC HLA SNPs e.g. rs61447909 from Adolphe 2021 PMID: 33549134

b. BCC: chr8 rs2241261 is in LD r^2 0.5 with rs2241260 in Adolphe 2021 (see their cojo analysis in sup table 2). chr12 rs2853562 is in LD $r^2 \sim 1$ with Adolphe 2021 chr12 rs739837.

c. BCC:chr10 rs12767525 is reported in Liyanage 2019 PMID 31174203 and in LD r^2 0.8 with rs35202367 from Adolphe 2021

d. BCC:Chr 3 rs9858861, chr10 rs709811, chr12 rs772920 are reported in Seviiri 2022 PMID 36496446

e. BCC:chr11 rs112568268 is also in LD $r^2 \sim 0.16$ with rs73008229, a SNP in Seviiri 2022 PMID 36496446. However, it is interesting to hypothesise both rs112568268 from this analysis and rs73008229 are associated with BCC due to LD with rs180516, a non-synonymous SNP in ATM previously associated with cutaneous melanoma. Does the cojo analysis or variant prioritisation analysis shed any light on this?

f. BCC: The four SNPs on chr20 from 31-36 Mb (rs403598, rs4911466, rs75653149, rs7309491) are all in LD with the lead SNP at ASIP, rs6059655 e.g. using UK Biobank samples (one of the larger contributing sets) to calculate LD r^2 between these SNPs and rs6059655 shows they are all between r^2 0.25 to 0.75. The LD across the ASIP region is strong and long-range, extending over multiple megabases. Further, Open Targets etc reports that these SNPs are all associated with the same phenotypes as the ASIP rs6059655 SNP (e.g. tanning). Does the cojo analysis or variant prioritisation analysis shed any light on this?

g. For the SCC results - chr3 rs2049218 (LPP) was $P < 5e-8$ for SCC in Liyanage 2019 and there is the same issue of chr20 SNPs in LD with rs6059655 listed as novel.

5. Results line 66 onwards: Many loci are heterogeneous in terms of effect size; with ~ 12 of the novel BCC loci not significant ($P > 5e-8$) under a random effects model. While this doesn't invalidate these loci this should be discussed as the heterogeneity may be biologically relevant (e.g. differences across cohorts in ascertainment, phenotype definition, age, risk factors etc.), due to a strong effect in one

study, or potentially due to biases in one study etc. Forrest plots of new loci would be helpful to interpret this.

6. Results line 66 onwards. The GIF for the BCC GWAS meta-analysis 1.242. This may just be due to the study being large and well-powered, so reporting the LDSC intercepts as well as GIF, and the ratio of the two should clarify if the inflation in test statistics is due to actual biases e.g. stratification. If the LD score intercept is also high this should be discussed/addressed.

7. Results line 69 - The lead SNPs in a meta-analysis should have correlated effect sizes otherwise they wouldn't be the lead SNPs. The correlation between effect sizes at lead SNPs in an independent cohort (e.g. not used to identify the SNPs) is more informative. Failing that, the correlation between lead SNPs identified in a leave-one-out meta-analysis and the excluded set could be used. However, heterogeneity metrics like I^2 , Q , or the random effects p-value are more informative with respect to the consistency of effect sizes within meta-analyses (e.g. see point 5).

a. As it is I am not sure what this section is adding.

b. If the authors wish to retain this figure the lead SNPs should be selected excluding the comparison set.

c. The list of included SNPs should also exclude correlated SNPs (e.g. around ASIP) as they may make the correlations appear stronger than they are.

8. Related to 7, sup figure 10 should be the correlation in effect sizes from lead SNPs identified without the Hispanic/Latino GWAS being included (e.g. lead SNPs from sup table 2, not sup table 4 - the legend of sup table 10 suggests the latter). Otherwise, the SNPs chosen are again the ones most consistent across the combined sets. However, this figure, if constructed correctly excluding the Hispanic/Latino GWAS from the discovery set is very relevant and should be retained

a. This plot could potentially be better as a main text figure?

b. Though more useful and interesting would be reporting how well a EUR-derived PRS predicts BCC/SCC risk (e.g. AUC) in the independent Hispanic/Latino population. Including this would greatly strengthen this manuscript.

c. One of the contributing Hispanic/Latino GWAS has a very small case count ($N = 19$) so correlations should be displayed separately for each GWAS rather than for the meta-analysis alone.

9. Results line 70-76 - is sup table 3 the results for the conditional analysis of all loci or just showing the secondary signals? If the latter the full results should be displayed; this is more informative than just secondary signals as it helps interpret the GWAS meta-analysis as a whole.

10. It is not clear how the reported/listed genes for SNPs/loci are derived.

a. Intro line 74 - for chr5p15.33 while the plausible candidate gene is TERT the situation is unclear; chr5 rs410805 reported here is only r^2 0.09 with telomere SNPs such as rs7705526 (e.g. Codd 2021 PMID: 34611362). Still, this is worth addressing/discussing.

b. Intro line 74: The gene for chr16q24.3 is MC1R - the lead SNP reported in the sup is rs12931267, which is in LD (r^2 0.9) with rs1805007, a non-synonymous SNP in MC1R associated with red hair/fair skin.

c. chr20 rs6059655 has MIR4755-RALY listed as the gene, but this SNP is an eQTL for ASIP in the skin (e.g. See Open Targets, or Hysi et al 2018 PMID: 29662168). Given the functional relationship between ASIP and MC1R, and the phenotypes this SNP is associated with, ASIP is more likely the target gene than MIR4755-RALY.

d. Related, in the footnotes for tables, and figures (e.g. sup figure 2) it is unclear how the listed genes were chosen.

e. In the methods please clearly detail how the listed genes associated with the lead SNPs were identified.

f. Also see point 13

11. Results line 105-106. My understanding is that the input GWAS/cohorts are the same for both BCC

and SCC (GERA, UKB, MGB, and 23andMe). So the only differences should be different phenotypes applied to the same underlying GWAS data. If so, how are only 101/107 lead SNPs from BCC present in the SCC analysis? Wouldn't the cleaning and excluded/retained SNPs etc be identical? Are there no proxies for the 6 missing SNPs?

12. Results page 111 - please report the pairwise genetic correlation (R_g) by cohort and by BCC/SCC (e.g. 23andMe SCC vs UKBB BCC etc.). This is relevant to the discussion lines 199-212 where the differences in phenotype classification across cohorts are discussed; in this case the R_g between cohorts (and potentially the pairwise correlation of effect sizes in leave-one-out analyses, point 7) is informative and should be referenced here.

13. Discussion - lines 151 - 169. Related to point 10, what is the functional evidence linking the chosen SNPs/loci and the discussed genes? E.g. at rs6700380 only PTPN14 is discussed but open targets report that PCHI-C data links this SNP to both PTPN14 and CENPF. rs6741117 is an eQTL for GRHL1 in the skin - is this why this gene was chosen? rs111304635 is an eQTL for NDFIP1 but not SPRY4-AS1, but SPRY4-AS1 is the topic of the discussed.

a. Please state how/why discussed genes were selected as the likely candidate gene underlying each association. If the gene is chosen based solely on proximity this should be noted.

b. Minor point - please also note the chr/SNP as well as the gene when discussing a locus to help cross-referencing SNPs/results with tables.

14. Discussion lines 194 - 198 - is the modification of melanin biosynthesis by drugs a viable therapy or treatment for BCC? The evidence strongly supports that melanin biosynthesis is relevant as a risk factor (low pigmentation increases DNA damage from UV) so restoring or increasing melanin biosynthesis seems unlikely to be able to treat BCC or SCC once developed. Further, a discussion of the viability of druggable modification of this pathway as a preventative approach (rather than therapy) would still need to be contrasted to existing established interventions to reduce UV damage such as sun avoidance, sun protective behaviour, sunscreen etc. Further discussion of this topic should address that there is literature on agonists for MC1R e.g. Afamelanotide.

15. Methods line 271-274: for GERA the controls were all non-cases. Does this mean the controls for the SCC GWAS control included BCC cases and vice versa? Or were any skin cancers excluded from controls? This could be clearer and may be an issue if the BCC cases were a relatively large proportion of the controls for the SCC GWAS and vice versa; if so this should be addressed.

16. Methods - BCC and SCC cannot be identified by ICD codes alone in UKBB as their codes only have 3 digits e.g. C447. As per PMID 33549134 BCC and SCC require histology data as well (e.g. matching field 40011). I assume histology data was used, but the exact process should be noted in the methods. If not done this would need to be corrected/addressed.

17. The reporting of GWAS cleaning/QC is sparse and inconsistent across the sets. For example, some data on cleaning missing genotypes are reported for GERA but not for other QC filters, and nothing is reported for sets such as MGB or UKBB.

a. Please report standard GWAS cleaning/filtering steps used for all sets e.g. missing SNP/individual threshold, the identification of ancestry outliers, etc.

b. Sup Table 1/methods Why is the control count for the SCC GWAS in UKBB 40k less than the BCC GWAS? How did filtering/cleaning differ?

c. How were related samples identified, removed, or adjusted for in the analyses?

d. How was UKB analysed - a mixed-model approach such as SAIGE or REGENIE?

e. Were any filters applied to imputed SNPs prior to meta-analysis e.g. MAF or RSQ? If so please report them, if not appropriate threshold should be applied e.g. MAF 0.01 or 0.001, $rsq > 0.5$.

Minor points

18. Sup figure 3; please order Locuszoom plots by chr:bp position; panel 1 plots are out of order

19. Results line 138 + methods - please detail in the methods how Open Targets was used e.g. if any filters were applied to select eQTLs, drug targets

20. BCC chr5/rs76748680 doesn't match any variants in Gnomad. dbSNP reports rs76748680 has merged into rs3212625 which is a complex multi-T deletion/insertion. Gnomad reports rs1484215266 at that position which also has G/GT alleles, though, and that is the rsID ID Open Targets uses for that SNP as well. They are likely the same variant as a T/- SNP is consistent with G/GT but swapping rs76748680 for rs1484215266 might be more consistent across databases.

21. Discussion 185-187 - mendelian randomisation of vitamin D levels suggests that higher levels of vitamin D may be a causal risk for BCC PMID: 33431812 which is relevant when proposing vitamin D receptors may be valid drug targets.

22. Discussion 190-193. Are these drug targets newly identified in this study?

23. Methods line 288-291 and 295-298 - the sections referring to preferring variants from one reference panel over the other is unclear and could be rephrased for clarity.

24. Methods line 314 -please note the specific type of meta-analysis e.g. inverse variance weighted meta-analysis of log(OR) effect sizes etc.

25. Ref 66 is for LDSC regression; the reference for bivariate R_g is PMID: 26414676 and should also be cited.

Reviewer #2 (Remarks to the Author):

Summary:

The manuscript entitled "Multi-ancestry genome-wide meta-analysis identifies novel basal cell carcinoma (BCC) loci and shared genetic effects with squamous cell carcinoma" by Choquet et al reports large-scale meta-analysis of BCC in individuals with European ancestry and Hispanic/Latinos. A total of 50 531 participants with BCC and 762 234 control participants from four cohorts were available. The authors identified 112 BCC-associated loci, of which 46 were novel, and fine-mapped these associations. The pigment gene SLC45A2 was associated with BCC in Hispanic/Latinos. In addition, the loci associated with BCC were investigated in 17,181 cutaneous squamous cell carcinoma (SCC) cases and 713,994 controls of European ancestry with 31 SNPs showing evidence of association. Finally, the authors identified 7 novel genome-wide loci associated with cSCC. The authors' major claim is that the study provides new insights into the genetic basis of BCC and cSCC susceptibility and that they uncovered potential new druggable genes.

Pathways identified were relevant to the phenotypes being investigated. The authors comprehensively discuss the limitations of the study, including that the methodology to identify cases differed between the cohorts and included self-identification in the case of one of the cohorts. It is encouraging to see the inclusion of ethnicities other than Europeans in investigations of this nature. As the authors admit, functional verification of the identified associations should be the next step, so the claim that druggable targets were identified is perhaps premature. Even so, this is an interesting study which produced convincing findings given the large sample sizes available.

Minor Comments:

1. Are there known differences in BCC incidence between different ethnicities? Please add this information to the introduction.

Reviewers' comments:

Reviewer #1 (Remarks to the Author):

The authors report on a large GWAS meta-analysis of BCC and SCC. Importantly they have extended their analysis to include individuals with Hispanic/Latino ancestry.

We thank the reviewer for the constructive review.

Major comments

1. Introduction line 40 and/or 43: Please report the twin or family-based heritability estimates for BCC and SCC to contextualise the SNP-h² estimates e.g. Mucci 2016 PMID: 26746459

We have now cited the paper by Mucci et al (JAMA 2016) in the **Introduction**, as follows:

“Keratinocyte carcinoma has a moderate genetic component with a twin or family-based heritability estimate of 43.0% and array-heritability estimates up to 17.0% for BCC.”

2. Intro line 41: A recent multi-trait GWAS of BCC and SCC is not cited PMID: 36496446. The overlap in terms of findings with that work and this one should be discussed.

We would like to clarify that at the time of our manuscript submission (December 2022), the paper conducted by Seviiri and colleagues (Nat Commun. 2022 - PMID: 36496446) was not published yet. However, we agree with the reviewer that this is an important study, and we have now cited this paper in the **Introduction**, as follows:

“Previously published genome-wide association studies (GWAS) (Seviiri et al. Nat Comms. 2022), mainly conducted in European descent populations, have reported more than 78 loci associated with BCC, explaining up to 11.0% of BCC heritability.”

We have also discussed the findings overlap between the current study and the recent study conducted by Seviiri and colleagues in the **Discussion**, as follows:

“We recognize some potential limitations of our study. (...) Second, in order to maximize power to detect novel BCC and SCC risk loci, we did not separate our sample into a separate discovery and replication set. However, our results replicated most of previous known loci, especially from a recent large multi-trait genetic analysis that reported 78 risk loci for BCC (Seviiri et al. Nat Comms. 2022).”

3. Intro lines 49-51. Please indicate which GWAS have been previously reported or are new and the proportion of new samples relative to previous publications.

We have now indicated in the **Introduction**, the data sets that have been previously used in GWAS of BCC studies, as below:

“Here, we present a large GWA meta-analysis of BCC, including 802,297 individuals of European ancestry (49,905 BCC cases) from the Genetic Epidemiology Research on Adult Health and Aging (GERA) cohort, the Mass-General Brigham (MGB) Biobank cohort, the UK Biobank (UKB) cohort, and the 23andMe, Inc. research cohort (Supplementary Data 1). As a note, the GWAS data set from the 23andMe research cohort consisted of 12,945 BCC cases and 274,252 controls of European ancestry and was previously used in a discovery GWAS (Chahal et al. Nat Comms. 2016) and more recently in BCC GWA meta-analyses (Liyanage et al. HMG 2019; Seviiri et al. Nat Comms. 2022). Similarly, the UKB data set has been previously included in recent BCC GWA meta-analyses (Liyanage et al. HMG 2019; Seviiri et al. Nat Comms. 2022).”

4. Results line 66 - some of the novel loci are reported in (or are in LD with previously reported loci) in Adolphe 2021 PMID: 33549134, Liyanage 2019 PMID 31174203, Seviiri 2022 PMID 36496446 etc. Methods line 318-19 suggests only distance was used to identify new loci. This must also include an appropriate, strong, filter for LD as well e.g. $r^2 < 0.05$ and/or an approach like GCTA cojo, as used later in the methods, to determine which loci are independent of other lead SNPs. The aim of reporting novel loci should be to identify new biology/genes/pathways reliably associated with the risk of SCC or BCC, and SNPs whose association is driven by previously identified regions aren't achieving this goal. While I have highlighted specific examples below, both LD and distance to prior reports should be considered for all findings:

This is an important point raised by the reviewer. We have now rigorously compared our BCC-associated loci with previously reported ones from more recent studies (Adolphe C et al. *Genome Medicine* 2021; and Seviiri M et al. *Nat Commun.* 2022) - in addition to our original comparison with BCC loci identified in the previous studies (Chahal HS et al. *Nat Commun.* 2016; Liyanage UE et al. *HMG* 2019).

To define the ‘novel loci’ in the current study and to make sure that the identified SNPs were independent from previously reported ones, we have now considered linkage disequilibrium (LD) pattern using LDlink (<https://ldlink.nih.gov/>) (Machiela MJ, Chanock SJ. *Bioinformatics.* 2015) in addition to genomic distance.

We have now reflected this change in the **Methods** section, as follows:

*“For each locus, we defined the top SNP as the most significant variant within a 2 Mb window. Novel loci were defined as those that were located over 1 Mb apart from any previously reported locus, (...) Loci were also defined as novel if the identified lead SNPs were not in linkage disequilibrium (LD) with previously reported SNPs using LDlink tool (Machiela MJ, Chanock SJ. *Bioinformatics.* 2015).”*

Reference: Machiela MJ, Chanock SJ. LDlink: a web-based application for exploring population-specific haplotype structure and linking correlated alleles of possible functional variants. *Bioinformatics.* 2015

After those considerations (i.e. comparison with more recent BCC genetic studies and considering LD pattern between newly and previously reported SNPs), we have now identified 37 novel BCC-associated loci (instead of 41 novel loci, originally). We have reflected this updated number of novel BCC loci throughout the manuscript.

a. BCC: chr6/rs29243. LD is complex and far-reaching in the HLA region. Do the cojo results or variant prioritisation (sup table 8) support this variant as being independent of other known BCC HLA SNPs e.g. rs61447909 from Adolphe 2021 PMID: 33549134

We have double checked and those 2 SNPs (rs29243 at *GABBR1* on chr 3 p22.1 and rs61447909 on chr 3 p21.33) seem independent as: 1) they are relatively far from each other (1.81Mb apart); and 2) they are in linkage equilibrium ($R^2 = 0.046$ and $D' = 0.5299$). For those reasons, we still consider rs29243 at *GABBR1* as a novel BCC signal.

b. BCC: chr8 rs2241261 is in LD r^2 0.5 with rs2241260 in Adolphe 2021 (see their cojo analysis in sup table 2). chr12 rs2853562 is in LD $r^2 \sim 1$ with Adolphe 2021 chr12 rs739837.

As those lead SNPs at *RHOBTB2* (i.e., rs2241261 and rs2241260) are relatively close to each other (754 bp apart) and are moderately correlated in European-ancestry populations ($R^2 = 0.51$ and $D' = 1.0$), we no longer emphasize this *RHOBTB2* locus as a novel BCC locus in our study, and we have deleted it from Table 1 “*Novel BCC loci identified in the European ancestry and/or the multi-ancestry GWA meta-analyses*”.

Similarly, as lead SNP rs2853562 at *VDR* is relatively close to previously reported rs739837 (1,835 bp apart) and are strongly correlated in European-ancestry populations ($R^2 = 0.99$ and $D' = 1.0$), we no longer emphasize this *VDR* locus as a novel BCC locus in our study, and we have deleted it from Table 1 “*Novel BCC loci identified in the European ancestry and/or the multi-ancestry GWA meta-analyses*”.

c. BCC:chr10 rs12767525 is reported in Liyanage 2019 PMID 31174203 and in LD r^2 0.8 with rs35202367 from Adolphe 2021

We thank the reviewer for catching this. We no longer emphasize this *GATA3-SFTA1P* locus as a novel BCC locus in our study, and we have deleted it from Table 1 “*Novel BCC loci identified in the European ancestry and/or the multi-ancestry GWA meta-analyses*”.

d. BCC:Chr 3 rs9858861, chr10 rs709811, chr12 rs772920 are reported in Seviiri 2022 PMID 36496446

As above-mentioned, we would like to emphasize that the *Nat Commun.* 2022 paper by Mathias Seviiri et al. was published after we submitted our manuscript. For this reason, in our original manuscript, we considered those three loci (i.e., *PA2G4P4-LEKRI*, *EMX2-RAB11FIP2*, and *RAB5B*) as novel BCC loci. We have now deleted those three loci from Table 1 “*Novel BCC loci identified in the European ancestry and/or the multi-ancestry GWA meta-analyses*”.

e. BCC:chr11 rs112568268 is also in LD $r^2 \sim 0.16$ with rs73008229, a SNP in Seviiri 2022 PMID 36496446. However, it is interesting to hypothesise both rs112568268 from this analysis and rs73008229 are associated with BCC due to LD with rs180516, a nonsynonymous SNP in ATM previously associated with cutaneous melanoma. Does the cojo analysis or variant prioritisation analysis shed any light on this?

We found that lead SNP rs112568268 at *C11orf65* is relatively close to previously reported rs73008229 (110.9 kb apart) and are correlated in European-ancestry populations ($R^2 = 0.16$ and $D' = 1.0$), we no longer emphasize this *C11orf65* locus as a novel BCC locus in our study, and we have deleted it from Table 1 “*Novel BCC loci identified in the European ancestry and/or the multi-ancestry GWA meta-analyses*”.

f. BCC: The four SNPs on chr20 from 31-36 Mb (rs403598, rs4911466, rs75653149, rs7309491) are all in LD with the lead SNP at ASIP, rs6059655 e.g. using UK Biobank samples (one of the larger contributing sets) to calculate LD r^2 between these SNPs and rs6059655 shows they are all between r^2 0.25 to 0.75. The LD across the ASIP region is strong and long-range, extending over multiple megabases. Further, Open Targets etc reports that these SNPs are all associated with the same phenotypes as the ASIP rs6059655 SNP (e.g. tanning). Does the cojo analysis or variant prioritisation analysis shed any light on this?

We agree with the reviewer that the LD across the ASIP region is strong and long-range, extending over multiple megabases. To reflect this point, we have added some text in the **Methods** section, as follows:

“Novel loci were defined as those that were located over 1 Mb apart from any previously reported locus, except for the genomic region 20q11 (around ASIP) where strong and long-range linkage disequilibrium (LD) has been observed and for which an extended window of 2.5 Mb was used.”

We have also found moderate to strong LD between lead SNP at *RALY/ASIP* rs6059655 and rs4911466 ($R^2 = 0.79$ and $D' = 0.90$); rs6059655 and rs75653149 ($R^2 = 0.47$ and $D' = 0.72$); and rs6059655 and rs7309491 ($R^2 = 0.26$ and $D' = 0.60$). For those reasons, we no longer emphasize those 3 loci at 2q11 as novel BCC loci in our study, and we have deleted them from Table 1 “*Novel BCC loci identified in the European ancestry and/or the multi-ancestry GWA meta-analyses*”. However, we found that rs6059655 and rs403598 were relatively far from each other (2.44 Mb apart); and 2) they are in linkage equilibrium ($R^2 = 0.0002$ and $D' = 1.0$). For those reasons, we still consider rs403598 at *BPIFB6-BPIFB3* as a novel BCC signal.

g. For the SCC results - chr3 rs2049218 (LPP) was $P < 5e-8$ for SCC in Liyanage 2019 and there is the same issue of chr20 SNPs in LD with rs6059655 listed as novel.

We thank the reviewer for catching this. We no longer emphasize these *LPP* and *RALY/ASIP* loci as novel SCC loci in our manuscript.

5. Results line 66 onwards: Many loci are heterogeneous in terms of effect size; with ~12 of the novel BCC loci not significant ($P > 5e-8$) under a random effects model. While this doesn't invalidate these loci this should be discussed as the heterogeneity may be biologically relevant (e.g. differences across cohorts in ascertainment, phenotype definition, age, risk factors etc.), due to a strong effect in one study, or potentially due to biases in one study etc. Forrest plots of new loci would be helpful to interpret this.

We have now generated the Forest plots of new BCC loci and reported those in a Supplementary Figure (Supplementary Fig. 4). The effect estimates of the lead SNPs were consistent across the 4 studies (Table 1 and Supplementary Fig. 4), except for two SNPs: rs112108851 at *ZAN* and rs34302850 at *INPP5E* for which we observed an inconsistent direction of effects in the MGB cohort compared to the three other cohorts (GERA, UKB, and 23andMe). However, we would like to emphasize that we did not detect significant heterogeneity among different cohorts for any of the novel BCC loci identified in the European ancestry GWAS meta-analysis. We have now acknowledged this point and discussed it in the manuscript, as follows:

In the **Results**: *“The effect estimates of the lead SNPs were consistent across the 4 studies (Table 1 and Supplementary Fig. 4), except for two SNPs: rs112108851 at ZAN and rs34302850 at INPP5E”.*

In the **Discussion**: *“We recognize some potential limitations of our study. First, it is important to note phenotypic differences for BCC between the 4 study cohorts. While in the GERA cohort, BCC cases were identified from electronic pathology records using a validated SNOMED code-based algorithm, in the MGB and UK Biobank,*

BCC cases were identified based on International Classification of Disease, Ninth (ICD-9) and/or Tenth (ICD-10) diagnosis codes, and in 23andMe research cohort, participants self-reported a history of BCC. This could have led to misclassification of BCC phenotype. However, our meta-analysis combining GERA, MGB, UKB, and 23andMe results showed consistency of the SNPs effect estimates between cohorts, except for two SNPs rs112108851 at ZAN and rs34302850 at INPP5E for which we observed an inconsistent direction of effects in MGB compared to the other cohorts.

6. Results line 66 onwards. The GIF for the BCC GWAS meta-analysis 1.242. This may just be due to the study being large and well-powered, so reporting the LDSC intercepts as well as GIF, and the ratio of the two should clarify if the inflation in test statistics is due to actual biases e.g. stratification. If the LD score intercept is also high this should be discussed/addressed.

Because of the large sample size, we obtained a genomic inflation factor lambda (λ) of 1.242 for the European ancestry GWA meta-analysis of BCC, which is reasonable for a dichotomous trait with polygenic inheritance with a sample size this large (Yang, J. et al. Eur. J. Hum. Genet. 2011). Since λ scales with sample size, some have found it informative to report λ_{1000} (Yang, J. et al. Eur. J. Hum. Genet. 2011), the inflation factor for an equivalent study of 1000 cases and 1000 controls, which can be calculated by rescaling λ , as below:

$$\lambda_{1000} = 1 + (\lambda_{\text{obs}} - 1) * (1/n_{\text{cases(obs)}} + 1/n_{\text{controls(obs)}}) / (1/n_{\text{cases(1000)}} + 1/n_{\text{controls(1000)}})$$

, where $n_{\text{cases(obs)}}$ and $n_{\text{controls(obs)}}$ are the study sample size for cases and controls, respectively, and $n_{\text{cases(1000)}}$ and $n_{\text{controls(1000)}}$ are the target sample size (1000).

So, we have calculated the lambda 1000 for the European ancestry BCC sample, as follows:

$$\lambda_{1000} = 1 + (1.242 - 1) * (1/49,905 + 1/752,392) / (1/1000 + 1/1000)$$

$$\lambda_{1000} = 1.0026$$

We obtained the value of 1.0026 for λ_{1000} , which is reasonable for a genomic inflation factor under the assumption of polygenic inheritance.

Similarly, we have also calculated the lambda 1000 for the European ancestry GWA meta-analysis of SCC, and we have added this information in the **Results** section, and we have also added a reference (Yang, J. et al. Eur. J. Hum. Genet. 2011) to justify the initial lambda values of 1.242 and 1.122, for BCC and SCC meta-analysis, respectively, as below:

“In the GWA meta-analysis combining results from four cohorts (i.e., GERA, UKB, MGB, and 23andMe research cohort) and consisting of 49,905 BCC cases and 752,392 controls of European ancestry, we identified 107 loci associated with BCC ($P < 5 \times 10^{-8}$; $\lambda = 1.242$ and $\lambda_{1000} = 1.003$, which is reasonable for a sample of this size under the assumption of polygenic inheritance) (...) Our genome-wide analysis of SCC revealed 26 genome-wide significant loci ($\lambda = 1.122$ and $\lambda_{1000} = 1.004$, which is reasonable for a sample of this size under the assumption of polygenic inheritance)”.

Reference: Yang, J. et al. Genomic inflation factors under polygenic inheritance. Eur. J. Hum. Genet. 19, 807–812 (2011)

As suggested by the reviewer, we have also assessed the LDSC intercept (Bulik-Sullivan, B. K. et al. Nat. Genet. 2015), which showed no substantial inflation for both BCC (LDSC intercept = 1.025, 95% CI = 1.006–1.045) and SCC (LDSC intercept = 1.055, 95% CI = 1.034–1.074).

7. Results line 69 - The lead SNPs in a meta-analysis should have correlated effect sizes otherwise they wouldn't be the lead SNPs. The correlation between effect sizes at lead SNPs in an independent cohort (e.g. not used to identify the SNPs) is more informative. Failing that, the correlation between lead SNPs identified in a leave-one-out meta-analysis and the excluded set could be used. However, heterogeneity metrics like I2, Q, or the random effects p-value are more informative with respect to the consistency of effect sizes within meta-analyses (e.g. see point 5).

a. As it is I am not sure what this section is adding.

We agree with the reviewer, and we have now deleted this sentence of the **Results** and replaced it with the following statement:

~~“We observed a very high concordance of the effect estimates of the 41 lead SNPs at novel loci across the four studies (Pearson’s correlation coefficients between each pair of studies ranged from 0.90 to 0.97, $P \leq 3.90 \times 10^{-33}$) (Supplementary Fig. 4). The effect estimates of the lead SNPs were consistent across the 4 studies (Table 1 and Supplementary Fig. 4), except for two SNPs: rs112108851 at ZAN and rs34302850 at INPP5E.”~~

We have also deleted the original Supplementary Figure 4 entitled ‘Correlation of effect sizes across cohorts for the lead 107 BCC-associated lead SNPs identified in the European ancestry GWA meta-analysis of BCC’ and replaced it with Forest plots of new BCC loci.

We also agree with the reviewer that heterogeneity metrics like I², Q, or the random effects p-value are informative with respect to the consistency of effect sizes within meta-analyses, this is the reason why we reported those values in Table 1 and Supplementary Data 2. Importantly, we did not detect significant heterogeneity among different cohorts for any of the novel BCC loci identified in the European ancestry GWAS meta-analysis.

b. If the authors wish to retain this figure the lead SNPs should be selected excluding the comparison set.

As above-mentioned, we have now deleted the original Supplementary Figure 4 and replaced it by Forest plots of new BCC loci.

c. The list of included SNPs should also exclude correlated SNPs (e.g. around ASIP) as they may make the correlations appear stronger than they are.

We agree with the reviewer, and as mentioned in our answer to the comment #4.f. we no longer emphasize those 3 loci at 2q11 as novel BCC loci in our study, and we have deleted them from **Table 1** “*Novel BCC loci identified in the European ancestry and/or the multi-ancestry GWA meta-analyses*”. Further, we have added some text in the **Methods** section to justify this change as follows: “*Novel loci were defined as those that were located over 1 Mb apart from any previously reported locus, except for the genomic region 20q11 (around ASIP) where strong and long-range linkage disequilibrium (LD) has been observed and for which an extended window of 2.5 Mb was used.*”

8. Related to 7, sup figure 10 should be the correlation in effect sizes from lead SNPs identified without the Hispanic/Latino GWAS being included (e.g. lead SNPs from sup table 2, not sup table 4 - the legend of sup table 10 suggests the latter). Otherwise, the SNPs chosen are again the ones most consistent across the combined sets. However, this figure, if constructed correctly excluding the Hispanic/Latino GWAS from the discovery set is very relevant and should be retained

We would like to thank the reviewer for catching this incorrect information in the title of the original Supplementary Fig. 10 (now Supplementary Fig. 11). We confirm that this Figure reported the correlation in effect sizes from the lead SNPs (N=107) identified in the European ancestry GWA meta-analysis. We have now corrected the Supplementary Figure’s title as follows: “*Correlation of effect sizes across populations (European ancestry vs. Hispanic/Latino) for the lead 107 BCC-associated lead SNPs identified in the European ancestry GWA meta-analysis of BCC.*”

a. This plot could potentially be better as a main text figure?

While we agree with the reviewer about the importance of Supplementary Fig. 11, we would prefer to keep it in the Supplementary Information (as we have reached already the maximum number (n=5) of display items recommended by *Communications Biology*).

b. Though more useful and interesting would be reporting how well a EUR-derived PRS predicts BCC/SCC risk (e.g. AUC) in the independent Hispanic/Latino population. Including this would greatly strengthen this manuscript.

This is an excellent suggestion. We have now developed two polygenic risk scores (PRS) based on the BCC-associated loci identified in the European ancestry GWA meta-analysis and tested their capacity to predict BCC

in the GERA Hispanic/Latino sample. We have added some text throughout the manuscript to reflect those new results.

In the **Abstract**: *“We also reported a polygenic risk score derived from the European ancestry GWA meta-analysis that predicted BCC in Hispanic/Latinos (area under the curve (AUC) = 0.72).”*

In the **Results** section: *“European-derived PRS predicted BCC risk in Hispanic/Latinos. We first constructed two polygenic risk scores (PRSs) for BCC based on the GWAS summary statistics from the European ancestry GWA meta-analysis using two different P-value thresholds (i.e., $P < 5.0 \times 10^{-8}$ and $P < 1.0 \times 10^{-6}$) (see Methods). Each predictive model included a PRS, along with age and sex, and was predictive of BCC risk in the Hispanic/Latino sample from the GERA cohort with an area under the curve (AUC) value of 0.724 and 0.725, depending on the threshold of significance for the SNPs (P -value $< 5.0 \times 10^{-8}$ and $P < 1.0 \times 10^{-6}$, respectively) (Supplementary Fig. 8).”*

In the **Methods** section: *“Polygenic risk scores for BCC and prediction models. We constructed two PRSs for BCC based on the GWAS summary statistics from the European ancestry GWA meta-analysis, using two different P-value thresholds (i.e., $P < 5.0 \times 10^{-8}$ and $P < 1.0 \times 10^{-6}$). LD clumping was performed using a 10000 kb LD window and a r^2 cutoff of 0.005. The first PRS for BCC (PRS1) included 125 clumped independent SNPs at $P < 5.0 \times 10^{-8}$; and the second PRS (PRS2) included 176 clumped independent SNPs at $P < 1.0 \times 10^{-6}$. The two PRSs were calculated as a weighted sum of risk alleles by their estimated effect sizes. To assess the potential value of those two European-derived PRSs to predict BCC risk, regression-based models were tested in the Hispanic/Latino sample from the GERA cohort, which was not part of any of the analyses through which the genetic associations were identified. Each model included PRS (PRS1 or PRS2), age, and sex. The AUC were calculated and the receiver operating characteristic curves were drawn. The R programming language and software environment for statistical computing was used for calculating the two PRSs for BCC, as well as for conducting the logistic regression models (*‘glm’*) and for calculating AUC and generating the receiver operating characteristic curves (*‘pROC’*).”*

We have also added a Supplementary Figure (Supplementary Fig. 8) showing the receiver operating characteristic curves for BCC prediction in Hispanic/Latino individuals from GERA, based on the two PRSs derived from the European ancestry GWA meta-analysis of BCC.

c. One of the contributing Hispanic/Latino GWAS has a very small case count (N = 19) so correlations should be displayed separately for each GWAS rather than for the meta-analysis alone.

In addition to the Supplementary Fig. 11.a., we now provide the correlation of effect sizes across populations (European ancestry GERA only vs. Hispanic/Latino GERA only) for the lead 107 BCC-associated lead SNPs identified in the European ancestry GWA meta-analysis of BCC as a Supplementary Figure (Supplementary Fig. 11.b.).

9. Results line 70-76 - is sup table 3 the results for the conditional analysis of all loci or just showing the secondary signals? If the latter the full results should be displayed; this is more informative than just secondary signals as it helps interpret the GWAS meta-analysis as a whole.

We would like to clarify that in the original manuscript, we just reported independent signals within the 107 loci identified in the European ancestry GWA meta-analysis. As suggested by the reviewer, we have now displayed the full results of this COJO analysis in Supplementary Data 3. We have rephrased the **Results** section to reflect this point as follows:

“To identify additional and independent signals, we performed a multi-SNP-based conditional & joint association analysis (COJO), which revealed 40 additional independent SNPs within 24 loci. These include at 1) known BCC loci ALS2CR12 (chr2 q33.1), near CLPTMIL (chr5 p15.33), near FANCA (chr16 q24.3), and TGM3 (chr20 p13); 2) at the newly loci INCENP (chr11 q12.2), PIWIL4-AMOTL1 (chr11 q21), and FGF7-FAM227B (chr15 q21.2) identified in the European ancestry GWAS analysis; and 3) at an additional novel locus, ALKAL2 (chr2 p23.3) (Supplementary Data 3).”

10. It is not clear how the reported/listed genes for SNPs/loci are derived.

To clarify how the reported genes for each lead SNP/locus are derived, we have now modified the header title in Table 1 as “*Nearest gene(s)*” and added the following footnote after Table 1: “*We report the nearest gene(s) for each SNP; more complete annotation data (other genes within each locus) can be found in Supplementary Figure 3.*”

The “nearest gene” names were derived from the National Center for Biotechnology Information dbSNP tool (www.ncbi.nlm.nih.gov/snp/).

Also, we would like to emphasize that the other gene names at those novel BCC loci are reported in the LocusZoom plots (Supplementary Figure 3).

a. Intro line 74 - for chr5p15.33 while the plausible candidate gene is TERT the situation is unclear; chr5 rs410805 reported here is only r^2 0.09 with telomere SNPs such as rs7705526 (e.g. Codd 2021 PMID: 34611362). Still, this is worth addressing/discussing.

We would like to clarify that while the lead SNPs rs410805 (identified in the current study) and rs421284 (identified in a previous study) are intronic variants located within the *CLPTMIL* gene, the index SNP rs183126 (associated with BCC from COJO analyses) is located near *CLPTMIL*. We have now clarified this point in the **Results** section as follows:

“To identify independent signals within the 107 genomic regions identified in the European ancestry GWA meta-analysis, we performed a multi-SNP-based conditional & joint association analysis (COJO), which revealed 38 additional independent SNPs within 23 loci, including at known BCC loci (...) near CLPTMIL (chr5 p15.33) ...”.

b. Intro line 74: The gene for chr16q24.3 is MC1R - the lead SNP reported in the sup is rs12931267, which is in LD (r^2 0.9) with rs1805007, a non-synonymous SNP in MC1R associated with red hair/fair skin.

Similarly, while the lead SNPs rs12931267 (identified in the current study) is an intronic variant located within the *FANCA* gene, the index SNPs (associated with BCC from COJO analyses) are located near *FANCA*. We have now clarified this point in the Results section as follows:

“To identify independent signals within the 107 genomic regions identified in the European ancestry GWA meta-analysis, we performed a multi-SNP-based conditional & joint association analysis (COJO), which revealed 38 additional independent SNPs within 23 loci, including at known BCC loci (...) near FANCA (chr16 q24.3) ...”.

c. chr20 rs6059655 has MIR4755-RALY listed as the gene, but this SNP is an eQTL for ASIP in the skin (e.g. See Open Targets, or Hysi et al 2018 PMID: 29662168). Given the functional relationship between ASIP and MC1R, and the phenotypes this SNP is associated with, ASIP is more likely the target gene than MIR4755-RALY.

Similarly, the lead SNP rs6059655 (identified in the current study) is an intronic variant located within the *RALY* gene. We have now corrected the locus name (i.e. “*RALY*” instead of “*MIR4755-RALY*”) throughout the manuscript as, above-mentioned, we report the nearest gene for each lead SNP.

d. Related, in the footnotes for tables, and figures (e.g. sup figure 2) it is unclear how the listed genes were chosen.

We reported the nearest gene(s) for each SNP and we have now added the following footnote after Table 1: “*We report the nearest gene(s) for each SNP; more complete annotation data (other genes within each locus) can be found in Supplementary Figure 3.*”. Furthermore, more complete annotation data (other genes within each locus) can be found in Supplementary Figure 3 (Regional plots at each novel BCC locus).

e. In the methods please clearly detail how the listed genes associated with the lead SNPs were identified.

We have also added more details in the **Methods** section on how the listed genes associated with the lead SNPs were identified as follows:

“Novel loci were defined as those that were located over 1 Mb apart from any previously reported locus (...) Loci were also defined as novel if the identified lead SNPs were not in linkage disequilibrium (LD) with previously reported SNPs using LDlink tool. Locus names reported in the manuscript correspond to the nearest gene(s) for

[each lead SNP based on the National Center for Biotechnology Information dbSNP tool \(www.ncbi.nlm.nih.gov/snp/\).](http://www.ncbi.nlm.nih.gov/snp/)”

f. Also see point 13

11. Results line 105-106. My understanding is that the input GWAS/cohorts are the same for both BCC and SCC (GERA, UKB, MGB, and 23andMe). So the only differences should be different phenotypes applied to the same underlying GWAS data. If so, how are only 101/107 lead SNPs from BCC present in the SCC analysis? Wouldn't the cleaning and excluded/retained SNPs etc be identical? Are there no proxies for the 6 missing SNPs?

While the European ancestry GWA meta-analysis of BCC included a total of 9,057,132 genetic variants, the European ancestry GWA meta-analysis of SCC included a total of 8,225,106 genetic variants. As BCC and SCC are two different phenotypes, the number of individuals included in the 2 GWA analyses were different, so the minor allele frequency (MAF) for some variants may be slightly different between the 2 GWASs, especially for variants with low MAF. For this reason, some variants could have been excluded from the SCC GWAS based on the MAF.

As suggested by the reviewer, we have now looked for proxy of those 6 missing SNPs using LDlink (LDproxy web-based application) using European reference panels and a 2Mb window.

CHR	BP	Nearest Gene(s)	SNP ^a	Proxy from LDlink
2	173074802	DLX2-ITGA6	rs10646896	NA as ‘rs10646896 Variant is not in 1000G reference panel’
7	17134708	AGR3-AHR	rs117132860	rs62444531
13	113533651	ATP11A	rs1765871	rs1890202
19	4811956	FEM1A-TICAM1	rs11085101	rs34976338
19	50162909	IRF3	rs7251	rs10406006
20	49398786	PARD6B-BCAS4	rs62202836	rs113623356

We have now reported the associations with SCC with those proxy SNPs in Supplementary Data 7 and updated Figure 5.

12. Results page 111 - please report the pairwise genetic correlation (R_g) by cohort and by BCC/SCC (e.g. 23andMe SCC vs UKBB BCC etc.). This is relevant to the discussion lines 199-212 where the differences in phenotype classification across cohorts are discussed; in this case the R_g between cohorts (and potentially the pairwise correlation of effect sizes in leave-one-out analyses, point 7) is informative and should be referenced here.

As suggested by the reviewer, we have now conducted pairwise genetic correlation by cohort and by BCC/SCC and reported those new results throughout the manuscript. Please see below:

In the **Results** section: “*We also conducted genome-wide genetic correlation analyses between the 2 non-melanoma skin cancers (BCC and SCC) and by cohort using LD score regression. We found a high degree of genetic correlations between BCC and SCC and across the cohorts (Supplementary Figure 14 and Supplementary Data 8). For instance, BCC GWAS from GERA was highly correlated with SCC GWAS from 23andMe ($r_g = 0.98$; $se = 0.16$; $P\text{-value} = 1.81 \times 10^{-9}$), while the BCC GWAS from UKB was more moderately correlated with the SCC GWAS from GERA ($r_g = 0.70$; $se = 0.09$; $P\text{-value} = 1.22 \times 10^{-14}$).*”

In the **Discussion**: “*Further, we demonstrated substantial genetic correlation between BCC phenotypes across the different cohorts.*”

In the **Methods** section: “*Genetic correlation between BCC and SCC. A LD score regression was conducted, using the LDSC v1.0.1 command line tool, to estimate the genome-wide genetic correlations between BCC and SCC across cohorts. GWAS summary statistics of the European ancestry samples from the GERA, MGB, UKB, and 23andMe research cohorts for BCC and SCC were used as input data.*”

13. Discussion - lines 151 - 169. Related to point 10, what is the functional evidence linking the chosen SNPs/loci and the discussed genes? E.g. at rs6700380 only PTPN14 is discussed but open targets report that PChi-C data links this SNP to both PTPN14 and CENPF. rs6741117 is an eQTL for GRHL1 in the skin - is this why this gene was chosen? rs111304635 is an eQTL for NDFIP1 but not SPRY4-AS1, but SPRY4-AS1 is the topic of the discussed.

As above-mentioned (point 10), throughout the manuscript, we reported the nearest gene(s) for each SNP/locus. For instance, we identified rs6700380 which lies in an intron of *PTPN14*, in addition, our VEGAS2 gene-based association analysis prioritized *PTPN14* ($P=1.12 \times 10^{-4}$), suggesting that this gene could be the potential causal origin of the signal. Those reasons explain why we discussed about this gene in the Discussion, even if we agree with the reviewer that other genes at those BCC-associated loci could have been discussed based on functional evidence and prioritization.

a. Please state how/why discussed genes were selected as the likely candidate gene underlying each association. If the gene is chosen based solely on proximity this should be noted.

We have now stated in the **Discussion** the reasons why we discussed those specific genes as potential candidate genes underlying each association, as follows:

*“Interestingly, the VEGAS2 gene-based association analysis also prioritized those genes ($P=1.12 \times 10^{-4}$ for *PTPN14*; $P=5.43 \times 10^{-6}$ for *GRHL1*; and $P=7.76 \times 10^{-5}$ for *ITGA2*), suggesting that they could be the potential candidate genes underlying those associations. (...) Interestingly, the 12 variants that are more likely to explain the observed association at the BCC-associated *SPRY4-AS1* locus (i.e. included in the 95% credible set) were all located within *SPRY4-AS1* gene. This suggests that those variants may be the causal origin of the signal, and that *SPRY4-AS1* is more likely to be the candidate gene at this locus.”*

b. Minor point - please also note the chr/SNP as well as the gene when discussing a locus to help cross-referencing SNPs/results with tables.

This is an excellent suggestion. We have now specified the chromosome location and SNP in the **Discussion** to help the reader cross-referencing results across the manuscript as below:

*“Among the novel SNPs associated with BCC susceptibility in our European ancestry meta-analysis, we identified rs6700380 which lies in an intron of *PTPN14* (1q41). (...) We also identified an association between rs6741117 (which is an intronic variant of *GRHL1*, 2p25.1) with BCC. (...) We also identified an association between rs148421526 near *ITGA2* (2q37.1) with BCC. (...) We also identified an association between BCC and rs111304635, which lies in an intron of *SPRY4-AS1* (5q31.3).”*

14. Discussion lines 194 - 198 - is the modification of melanin biosynthesis by drugs a viable therapy or treatment for BCC? The evidence strongly supports that melanin biosynthesis is relevant as a risk factor (low pigmentation increases DNA damage from UV) so restoring or increasing melanin biosynthesis seems unlikely to be able to treat BCC or SCC once developed. Further, a discussion of the viability of druggable modification of this pathway as a preventative approach (rather than therapy) would still need to be contrasted to existing established interventions to reduce UV damage such as sun avoidance, sun protective behaviour, sunscreen etc. Further discussion of this topic should address that there is literature on agonists for MC1R e.g. Afamelanotide.

This is an important point raised by the reviewer, and we have added some text to reflect this point in the **Discussion** as follows:

*“Moreover, given that our pathway analyses highlighted the involvement of the melanin metabolic and biosynthetic process in BCC pathogenesis, drugs targeting genes (i.e. *TRPC1*, *SLC45A2*, *TYRP1*, *TYR*, *PMEL*, *DCT*, *OCA2*, *MYO5A*, *MC1R*, *CTNS*, *ASIP*, *DDT*, and *BCL2*) involved in those pathways could be potential therapies for BCC. Most of these genes are pigmentation-related genes (i.e. *SLC45A2*, *TYR*, *OCA2*, *MC1R*, and *ASIP*) and have a biological importance of lighter pigmentation in susceptibility to keratinocyte carcinoma. Even if agonists for some of these genes exist (e.g. Afamelanotide, a α -melanocyte-stimulating hormone (MSH) analogue that binds *MC1R*) to treat some skin diseases, because of the importance of sun exposure in keratinocyte*

carcinoma, the modification of the melanin biosynthesis by those drugs as a treatment for BCC is questionable. Thus, existing preventative interventions, such as sun avoidance, remain presumably the most appropriate approach to decrease keratinocyte carcinoma risk in those cases.”

15. Methods line 271-274: for GERA the controls were all non-cases. Does this mean the controls for the SCC GWAS control included BCC cases and vice versa? Or were any skin cancers excluded from controls? This could be clearer and may be an issue if the BCC cases were a relatively large proportion of the controls for the SCC GWAS and vice versa; if so this should be addressed.

We would like to clarify that our control groups for BCC and SCC included all the non-cases after excluding individuals who had a current or prior cancer registry history of cancer, or benign or in-situ tumors, and not having self-reported cancer at the time of enrollment. We have now specified this information in the **Methods** section as below:

“BCC cases were identified from electronic pathology records. After excluding individuals with any evidence of metastatic BCCs (SNOMED codes M80906, M809061, M809063, M80909, M809092, M809093, M80946, M809492, M809493, M80960), our control group included all the non-cases after excluding individuals who had a current or prior cancer registry history of cancer, or benign or in-situ tumors, or had a self-reported cancer at the time of enrollment, as previously done (Jue-Sheng Ong et al. Br J Cancer. 2018). Further, cSCC cases were defined as subjects whose pathology records were consistent with incident cSCC (invasive or in situ, excluding anogenital and mucosal SCCs); controls were subjects with no pathology records consistent with cSCC and similar to the BCC control group, we excluded individuals who had a current or prior cancer registry history of cancer, or benign or in-situ tumors, or had a self-reported cancer at the time of enrollment.”

16. Methods - BCC and SCC cannot be identified by ICD codes alone in UKBB as their codes only have 3 digits e.g. C447. As per PMID 33549134 BCC and SCC require histology data as well (e.g. matching field 40011). I assume histology data was used, but the exact process should be noted in the methods. If not done this would need to be corrected/addressed.

This is an important point raised by the reviewer. We have now added more details on the identification of BCC and SCC cases and controls for UKB participants in the **Methods**, as follows:

“In UKB, BCC or SCC cases were defined as participants with an ICD-9 or ICD-10 diagnosis code for BCC or SCC and based on histology data (e.g. field ID: 40011) (see full process in Supplementary Methods).”

In the **Supplementary Methods**, as follows:

“**BCC case definition:** have at least one ICD10 or ICD9 diagnosis

#ICD10 invasive

anything starting C44 (e.g. C447, C442 etc) in f.40006.0.0 PLUS 8090-8094 or 8097-98 in f.40011.0.0 or anything starting C44 (e.g. C447, C442 etc) in f.40006.1.0 PLUS 8090-8094 or 8097-98 in f.40011.1.0 or anything starting C44 (e.g. C447, C442 etc) in f.40006.2.0 PLUS 8090-8094 or 8097-98 in f.40011.2.0 or ...etc down to x=9 in f.40006.x.0 and f.40011.x.0

#ICD9 invasive

anything starting 173 (e.g. 1730, 1731 etc) in f.40013.0.0 PLUS 8090-8094 or 8097-98 in f.40011.0.0 or anything starting 173 (e.g. 1730, 1731 etc) in f.40013.1.0 PLUS 8090-8094 or 8097-98 in f.40011.1.0 or anything starting 173 (e.g. 1730, 1731 etc) in f.40013.2.0 PLUS 8090-8094 or 8097-98 in f.40011.2.0 or ...etc down to x=10 in f.40013.x.0 and f.40011.x.0

(...)

SCC case definition: have at least one ICD10 or ICD9 diagnosis

#ICD10 invasive

anything starting C44 (e.g. C447, C442 etc) in f.40006.0.0 PLUS 8070-8076 or 8078 in f.40011.0.0 or anything starting C44 (e.g. C447, C442 etc) in f.40006.1.0 PLUS 8070-8076 or 8078 in f.40011.1.0 or

anything starting C44 (e.g. C447, C442 etc) in f.40006.2.0 PLUS 8070-8076 or 8078 in f.40011.2.0 or ...etc down to x=9 in f.40006.x.0 and f.40011.x.0

#ICD9 invasive

anything starting 173 (e.g. 1730, 1731 etc) in f.40013.0.0 PLUS 8070-8076 or 8078 in f.40011.0.0 or anything starting 173 (e.g. 1730, 1731 etc) in f.40013.1.0 PLUS 8070-8076 or 8078 in f.40011.1.0 or anything starting 173 (e.g. 1730, 1731 etc) in f.40013.2.0 PLUS 8070-8076 or 8078 in f.40011.2.0 or ...etc down to x=10 in f.40013.x.0 and f.40011.x.0”

17. The reporting of GWAS cleaning/QC is sparse and inconsistent across the sets. For example, some data on cleaning missing genotypes are reported for GERA but not for other QC filters, and nothing is reported for sets such as MGB or UKBB.

a. Please report standard GWAS cleaning/filtering steps used for all sets e.g. missing SNP/individual threshold, the identification of ancestry outliers, etc.

We have now reported a more details on the genotyping, imputation, and quality control for each cohort, in particular for GERA and MGB, for which a GWAS of BCC has not been previously described, in the **Supplementary Information** (see **Supplementary Methods**).

b. Sup Table 1/methods Why is the control count for the SCC GWAS in UKBB 40k less than the BCC GWAS? How did filtering/cleaning differ?

Please see answer to comment #15.

c. How were related samples identified, removed, or adjusted for in the analyses?

In MGB, samples with evidence of familiar relationship (identity-by-descent [IBD] > 0.2) were excluded from the analyses and we have now provided this Information in the **Supplementary Information**. Further, in the 23andMe research cohort, a pairwise IBD analysis was also conducted to detect related individuals, and a maximal set of unrelated individuals was chosen for GWAS analysis, as previously described (Chahal et al. Nature Communications 2016).

d. How was UKB analysed - a mixed-model approach such as SAIGE or REGENIE?

We have now provided this information in the **Methods section**, as follows:

“GWAS and adjustment. In GERA, we first analyzed each ethnic group (non-Hispanic white, and Hispanic/Latino) separately. We ran a logistic regression of BCC and SCC and each SNP using PLINK v1.90 adjusting for age, sex, and ancestry principal components (PCs). (...) The GWAS analyses were also performed using a recent approach accounting for relatedness that fits a whole-genome regression model, implemented in REGENIEv2.0.2. The GWAS results generated using REGENIE were similar compared to those generated using PLINK. (...) In UKB, PLINK v1.90 was also used to conduct the GWA analysis, and age, sex, and genetic ancestry PCs were included as covariates.”

e. Were any filters applied to imputed SNPs prior to meta-analysis e.g. MAF or RSQ? If so please report them, if not appropriate threshold should be applied e.g. MAF 0.01 or 0.001, rsq > 0.5.

Detailed information on the genotyping, imputation, and quality control for each cohort is now reported in the **Supplementary Information** (see **Supplementary Methods**). For instance: *“In MGB, (...) QC steps of imputed data were conducted. Specifically, an info score >0.8 (high-quality imputed SNPs), SNP call rates >0.95, and MAF > 0.01 were retained in the association analyses.”*

Minor points

18. Sup figure 3; please order Locuszoom plots by chr:bp position; panel 1 plots are out of order

We have now re-ordered the Locus Zoom plots reported in Supplementary Figure 3 by chr:bp position.

19. Results line 138 + methods - please detail in the methods how Open Targets was used e.g. if any filters were applied to select eQTLs, drug targets

We now provide more details in the **Methods** about how Open Targets was used, as follows:

“Prioritization of drug targets. We used Open Targets (<http://genetics.opentargets.org>), to search for drugs currently in use or in clinical trials for treating other skin cancer or systemic diseases that target the BCC risk genes identified in the current study. Potential druggable genes were prioritized if they were close to the most significant variants.”

20. BCC chr5/rs76748680 doesn't match any variants in Gnomad. dbSNP reports rs76748680 has merged into rs3212625 which is a complex multi-T deletion/insertion. Gnomad reports rs1484215266 at that position which also has G/GT alleles, though, and that is the rsID ID Open Targets uses for that SNP as well. They are likely the same variant as a T/- SNP is consistent with G/GT but swapping rs76748680 for rs1484215266 might be more consistent across databases.

We have now changed the rs number for rs76748680 to rs1484215266 in **Table 1** and in the **Supplementary Data** for consistency across databases.

21. Discussion 185-187 - mendelian randomisation of vitamin D levels suggests that higher levels of vitamin D may be a causal risk for BCC PMID: 33431812 which is relevant when proposing vitamin D receptors may be valid drug targets.

We have now cited the paper by Jue-Sheng Ong et al (Nat Commun. 2021) in the **Discussion**, as follows:

“Our results also prioritize 14 genes within the identified BCC risk loci that are targeted by existing drugs, some of those are already in use/clinical trials for various cancer types, inflammatory skin diseases, or systemic diseases. For instance, VDR is targeted by calcitriol, a Vitamin D receptor agonist that is currently used to treat various diseases such as cancers, psoriasis, and atopic eczema. Interestingly, a Mendelian randomization study demonstrated a causal effect of vitamin D on BCC susceptibility, such as higher 25-hydroxyvitamin D levels increase risk of BCC. Consistently, clinical trials demonstrated a beneficial role of vitamin D compounds in the treatment of actinic keratosis and other clinical trials are in progress to evaluate calcitriol therapy for basal cell carcinoma.”

22. Discussion 190-193. Are these drug targets newly identified in this study?

We have now clarified in the **Discussion** that some of these genes (i.e., *CTLA4*) within BCC loci have been recently highlighted and prioritized as potential drug target:

*“Some other drug candidates targeting proteins encoded by BCC-associated loci are also currently under consideration in ongoing clinical trials for treating various cancer types (...) ipilimumab, a cytotoxic T-lymphocyte protein 4 inhibitor, targets *CTLA4*, and aldesleukin, an interleukin-2 receptor agonist, targets *IL2RA*; current clinical trials are testing therapies based on these drugs for Merkel cell skin cancer and other skin cancers. Our findings are consistent with a recent genetic study that also prioritized *CTLA4* as a potential drug target for BCC treatment.”*

23. Methods line 288-291 and 295-298 - the sections referring to preferring variants from one reference panel over the other is unclear and could be rephrased for clarity.

We have now rephrased the sentence referring to the reference panels for GERA and UKB cohorts in the **Supplementary Methods**, as follows:

“In GERA, imputation was done by array, and we additionally removed variants with call rates <90%. Genotypes were then pre-phased with Eagle v2.3.2, and then imputed with Minimac3, twice, using two reference panels, in a manner almost identical to that done in the UKB, and described previously. In detail, the two reference panels included (1) the EGA release of the Haplotype Reference Consortium (n=27,165; no indels) and (2) the 1000 Genomes Project Phase 3 (N=2,404; e.g., indels). The first reference panel (1) uses a larger sample size (including all the individuals in (2)) and should theoretically impute better, but it is only available at non-indels due to harmonization issues; the second reference panel (2) includes, e.g., indels, which are not present in the first panel, but at a smaller sample size. Thus, we imputed using both reference panels, but if a variant was in both (1) and (2), we used that which should have imputed better, ie., (1). As above-mentioned imputation in GERA

was almost identical to that of UKB as has been described (...) in the UKB, the two reference panels were (1) the full HRC (n=32,488; no indels) and (2) a merger of 1000 Genomes Phase 3 and the UK10K (n=6,285; indels)."

24. Methods line 314 -please note the specific type of meta-analysis e.g. inverse variance weighted meta-analysis of log(OR) effect sizes etc.

We have now specified the type of meta-analysis that we used in the **Methods** section, as follows:

"GWA meta-analyses. Fixed-effects meta-analyses (of BCC and SCC) were conducted to combine the results from GERA, MGB, UKB, and 23andMe research cohort using the inverse variance-based method, as implemented in PLINK."

25. Ref 66 is for LDSC regression; the reference for bivariate Rg is PMID: 26414676 and should also be cited.

We have now added this reference Brendan Bulik-Sullivan et al. Nat Genet. 2015 in the **Methods** section: *"Genetic correlation between BCC and SCC. A LD score regression was conducted, using the LDSC v1.0.1 command line tool, to estimate the genome-wide genetic correlation between BCC and SCC across cohorts."*

Reviewer #2 (Remarks to the Author):

Summary:

The manuscript entitled "Multi-ancestry genome-wide meta-analysis identifies novel basal cell carcinoma (BCC) loci and shared genetic effects with squamous cell carcinoma" by Choquet et al reports large-scale meta-analysis of BCC in individuals with European ancestry and Hispanic/Latinos. A total of 50 531 participants with BCC and 762 234 control participants from four cohorts were available. The authors identified 112 BCC-associated loci, of which 46 were novel, and fine-mapped these associations. The pigment gene SLC45A2 was associated with BCC in Hispanic/Latinos. In addition, the loci associated with BCC were investigated in 17,181 cutaneous squamous cell carcinoma (SCC) cases and 713,994 controls of European ancestry with 31 SNPs showing evidence of association. Finally, the authors identified 7 novel genome-wide loci associated with cSCC. The authors' major claim is that the study provides new insights into the genetic basis of BCC and cSCC susceptibility and that they uncovered potential new druggable genes.

Pathways identified were relevant to the phenotypes being investigated. The authors comprehensively discuss the limitations of the study, including that the methodology to identify cases differed between the cohorts and included self-identification in the case of one of the cohorts. It is encouraging to see the inclusion of ethnicities other than Europeans in investigations of this nature. As the authors admit, functional verification of the identified associations should be the next step, so the claim that druggable targets were identified is perhaps premature. Even so, this is an interesting study which produced convincing findings given the large sample sizes available.

We thank the reviewer for the positive feedback and the helpful comment.

We agree with the reviewer that the claim that druggable targets were identified was perhaps premature, and we have now de-emphasized this point throughout the manuscript, as follows:

In the **Abstract**: *"Our study findings provide new insights into the genetic basis of BCC and cSCC susceptibility and uncover potential new druggable genes."*

In the **Introduction**: *"The associated loci provide relevant pathways underlying BCC susceptibility and highlight candidate genes as potential BCC therapeutic candidates."*

In the **Results** section: *"Prioritization of candidate genes as potential drug targets for BCC. By leveraging multi-omics datasets (i.e., eQTL, chromatin interaction) using Open Targets, we prioritized 14 candidate genes that could be potential drug targets for BCC (Supplementary Data 11). These include AHR, CCND1, CTLA4, CTSS, FGF1, GABBR1, GLRA1, HLA-DRB5, ICOS, IL2RA, PIK3R1, TLR3, TP53, and VDR. We discuss the relevance of some of these candidate genes in the discussion section below."*

In the **Discussion**: *“Finally, we prioritized causal variants, candidate genes that could be potentially relevant drug targets for BCC, and pathways, using bioinformatic functional analyses. (...) Our results also prioritize 14 candidate genes within the identified BCC risk loci that are targeted by existing drugs, some of those are already in use/clinical trials for various cancer types, inflammatory skin diseases, or systemic diseases. (...) Further investigations are required to confirm the functionality of these genes in vivo and in vitro which may support the suitability of these potential new druggable genes as alternative treatments for BCC. (...) Finally, our fine-mapping and bioinformatic annotation analyses provide functional relevance for candidate genes involved in the pathogenesis of BCC, that could be potentially targeted for treating BCC.”*

Minor Comments:

1. Are there known differences in BCC incidence between different ethnicities? Please add this information to the introduction.

This is an important point raised by the reviewer. We have now added information related to differences in BCC incidence between different ethnicities in the Introduction, as follows:

“BCCs account for approximately 80% of skin cancers, and while skin cancer is far more common in fair-skinned individuals, Hispanic, Asian, and African ancestry individuals account for 4 to 5%, 2 to 4%, and 1 to 2% of skin cancer cases, respectively. Especially, BCCs are the most common skin cancer in European, Hispanic, and Asian ancestry individuals and the second most common in African ancestry individuals (approx. 1.8% of BCCs occur in African ancestry individuals).”

We have also cited the following references:

- Hogue L, et al. Basal Cell Carcinoma, Squamous Cell Carcinoma, and Cutaneous Melanoma in Skin of Color Patients. *Dermatol Clin.* 2019. PMID: 31466591
- Agbai ON, et al. Skin cancer and photoprotection in people of color: a review and recommendations for physicians and the public. *J Am Acad Dermatol.* 2014. PMID: 24485530
- Munjal A, Ferguson N. Skin Cancer in Skin of Color. *Dermatol Clin.* 2023. PMID: 37236716

Reviewers' comments:

Reviewer #1 (Remarks to the Author):

Overall the authors have done a good job in addressing my concerns but a few remain.

1. In response to comment 4 *"- we have now identified 37 novel BCC-associated loci (instead of 41 novel loci, originally). We have reflected this updated number of novel BCC loci throughout the manuscript."* However Table 1 has 36 SNPs/loci – please check the table and/or the in text total

2. In response to 6. *"As suggested by the reviewer, we have also assessed the LDSC intercept (Bulik-Sullivan, B. K. et al. Nat. Genet. 2015), which showed no substantial inflation for both BCC (LDSC intercept = 1.025, 95% CI = 1.006-1.045) and SCC (LDSC intercept = 1.055, 95% CI = 1.034– 1.074)." Please also report this in the manuscript – this is more reliable/relevant than the rescaling to an effective sample size of 1000.*

3. In response to 11. *"While the European ancestry GWA meta-analysis of BCC included a total of 9,057,132 genetic variants, the European ancestry GWA meta-analysis of SCC included a total of 8,225,106 genetic variants. As BCC and SCC are two different phenotypes, the number of individuals included in the 2 GWA analyses were different, so the minor allele frequency (MAF) for some variants may be slightly different between the 2 GWASs, especially for variants with low MAF. For this reason, some variants could have been excluded from the SCC GWAS based on the MAF."*

Could of is an odd word choice – were they? rs1765871, rs7251, and rs62202836 all have MAFs > 0.25; these are unlikely to be filtered due to a changing case count. Plus the chosen proxies will have to have a very similar freq – why did they pass QC but not the originals? Please detail the QC differences such that these SNPs were included/not included. This is relevant as these are the most strongly associated SNPs for BCC for these loci – it is odd they are dropped from the application of a different phenotype to the same data.

4. Response to 12 Rg results are interesting and well done but sup figure 15 is missing BCC_MGB and SCC_UKB; please include. Sup table 8 is missing results for BCC_MGB but does have SCC_UKB – so please include BCC_MGB

5. Response to Q15 *"BCC cases our control group included all the non-cases after excluding individuals who had a current or prior cancer registry history of cancer, or benign or in-situ tumors, or had a self-reported cancer at the time of enrollment, as previously done (Jue-Sheng Ong et al. Br J Cancer. 2018). Further, cSCC ... controls were subjects with no pathology records consistent with cSCC and similar to the BCC control group, we excluded individuals who had a current or prior cancer registry history of cancer, or benign or in-situ tumors, or had a self-reported cancer at the time of enrollment."*

If the same filtering was applied to both SCC and BCC controls – removing all people with cancer registry records, benign or in situ etc. - then the final control count will be the same for both SCC and BCC. Please provide sufficient detail in the methods to explain the difference in final control counts e.g. as seen in sup table 1. As noted later, if the control cleaning was the same it is still unclear why the UKBB SCC controls are 40k fewer than the BCC controls if the controls were filtered the same.

6. Response to Q17, which was about handling relatedness. The added text only details how relatedness was addressed in the MGB and 23andMe sets. It doesn't mention the two largest sets, GERA and UKBB, and reads like relatedness was not accounted for in either set. There is significant amount of relatedness in the UKBB dataset, which can lead to inflated p-values if not correctly accounted for. I am not sure of the status of GERA. Please explicitly note in the manuscript how relatedness was addressed in these populations.

If the plink analysis in GERA and UKB included related people please instead report the results using an appropriate methods like SAIGE – the response indicates this was done but no results are provided. Are reported loci still significant when correctly accounting for relatedness in GERA and UKB?

Reviewer #2 (Remarks to the Author):

I have no additional comments.

Reviewers' comments:

Reviewer #1 (Remarks to the Author):

Overall the authors have done a good job in addressing my concerns but a few remain.

We thank the reviewer for the positive feedback on our revised manuscript and the important additional comments.

1. In response to comment 4 “- we have now identified 37 novel BCC-associated loci (instead of 41 novel loci, originally). We have reflected this updated number of novel BCC loci throughout the manuscript.”. However Table 1 has 36 SNPs/loci – please check the table and/or the in text total

We thank the reviewer for catching this inconsistency. Importantly, to address the comment #6 about handling relatedness, we have now conducted the GWA analyses using REGENIE (which is a recent approach accounting for relatedness) in both GERA and UK Biobank cohorts and re-conducted the GWA meta-analyses. Accordingly, we have now updated the text in the manuscript and all the tables with those new results and double checked the consistency of the number of novel BCC loci reported in the main text and in Table 1.

2. In response to 6. “As suggested by the reviewer, we have also assessed the LDSC intercept (Bulik-Sullivan, B. K. et al. Nat. Genet. 2015), which showed no substantial inflation for both BCC (LDSC intercept = 1.025, 95% CI = 1.006-1.045) and SCC (LDSC intercept = 1.055, 95% CI = 1.034– 1.074).” Please also report this in the manuscript – this is more reliable/relevant than the rescaling to an effective sample size of 1000.

As suggested by the reviewer, we have now reported the LDSC intercept in the Results section of the manuscript: LDSC intercept = 1.064, 95% CI = 1.038-1.091 for the BCC analysis and LDSC intercept = 1.056, 95% CI = 1.032– 1.080 for the SCC analysis, showing no substantial inflation.

3. In response to 11. “While the European ancestry GWA meta-analysis of BCC included a total of 9,057,132 genetic variants, the European ancestry GWA meta-analysis of SCC included a total of 8,225,106 genetic variants. As BCC and SCC are two different phenotypes, the number of individuals included in the 2 GWA analyses were different, so the minor allele frequency (MAF) for some variants may be slightly different between the 2 GWASs, especially for variants with low MAF. For this reason, some variants could have been excluded from the SCC GWAS based on the MAF. “

Could of is an odd word choice – were they? rs1765871, rs7251, and rs62202836 all have MAFs > 0.25; these are unlikely to be filtered due to a changing case count. Plus the chosen proxies will have to have a very similar freq – why did they pass QC but not the originals? Please detail the QC differences such that these SNPs were included/not included. This is relevant as these are the most strongly associated SNPs for BCC for these loci – it is odd they are dropped from the application of a different phenotype to the same data.

We have now applied the same QC criteria for both BCC and SCC GWA analyses. We have now clarified this point in the Methods section as “Analyses were restricted to SNPs with minor allele frequency (MAF) >1% and an imputation quality score >0.5”. Thus, the number of genetic variants (n=9,552,974) included in the European ancestry GWAS meta-analysis of BCC is now very similar to the number of genetic variants (n=9,520,954) included in the European ancestry GWAS meta-analysis of SCC. Further, all the lead SNPs (N=116) identified in the European ancestry GWAS meta-analysis of BCC are now available in the European ancestry GWAS meta-analysis of SCC.

4. Response to 12 Rg results are interesting and well done but sup figure 15 is missing BCC MGB and SCC_UKB; please include. Sup table 8 is missing results for BCC_MGB but does have SCC_UKB – please include BCC_MGB

As above-mentioned, we have now conducted the GWA analyses using REGENIE in both GERA and UK Biobank cohorts. Using those new results, we have now obtained summary of genetic correlation results for the updated SCC_UKB and added those in Supplementary Table 8.

However, when looking at genetic correlations with BCC_MGB an error message was obtained. As this error message usually indicates that the input result for the concerned sample has very low heritability (h^2), we have calculated a univariate estimation of h^2 for MGB_BCC and found a h^2 estimate of 0.0026 (0.015) -which is very low-, suggesting not enough polygenic signals. We have added a note following Supplementary Figure 14 and Supplementary Data 9 to explain the potential reason of missing results for BCC_MGB for those genetic correlation analyses.

5. Response to Q15 “BCC cases our control group included all the non-cases after excluding individuals who had a current or prior cancer registry history of cancer, or benign or in-situ tumors, or had a self-reported cancer at the time of enrollment, as previously done (Jue-Sheng Ong et al. Br J Cancer. 2018). Further, cSCC ... controls were subjects with no pathology records consistent with cSCC and similar to the BCC control group, we excluded individuals who had a current or prior cancer registry history of cancer, or benign or in-situ tumors, or had a self-reported cancer at the time of enrollment.”

If the same filtering was applied to both SCC and BCC controls – removing all people with cancer registry records, benign or in situ etc. - then the final control count will be the same for both SCC and BCC. Please provide sufficient detail in the methods to explain the difference in final control counts e.g. as seen in sup table 1. As noted later, if the control cleaning was the same it is still unclear why the UKBB SCC controls are 40k fewer than the BCC controls if the controls were filtered the same.

We thank the reviewer for pointing this inconsistency to our attention. We have double checked and we confirm that we applied the same exclusion criteria for both UKB BCC and SCC control groups. We also confirm that the same number of controls (N= 388,893 individuals) were used for both the UKB BCC and UKB SCC GWA analyses. We have now corrected the number of controls for UKB in Supplementary Data 1 and in the manuscript.

6. Response to Q17, which was about handling relatedness. The added text only details how relatedness was addressed in the MGB and 23andMe sets. It doesn’t mention the two largest sets, GERA and UKBB, and reads like relatedness was not accounted for in either set. There is significant amount of relatedness in the UKBB dataset, which can lead to inflated p-values if not correctly accounted for. I am not sure of the status of GERA. Please explicitly note in the manuscript how relatedness was addressed in these populations.

If the plink analysis in GERA and UKB included related people please instead report the results using an appropriate methods like SAIGE – the response indicates this was done but no results are provided. Are reported loci still significant when correctly accounting for relatedness in GERA and UKB?

This is an important point raised by the reviewer. We have now conducted the GWA analyses using REGENIE in both GERA and UK Biobank cohorts. We then re-conducted the GWA meta-analyses by combining those new GERA and UKB results with 23andMe and MGB. We have updated our manuscript accordingly with those new results, as well as all the related manuscript items (all Figures, Tables, Supplementary Data and Figures).

Reviewer #2 (Remarks to the Author):

I have no additional comments.

REVIEWERS' COMMENTS:

Reviewer #1 (Remarks to the Author):

The authors have done an excellent job addressing all my concerns. I have no further comments.